# On quantum backpropagation, information reuse, and cheating measurement collapse

**Amira Abbas**
Google Quantum AI, Venice, California 90291, USA
University of KwaZulu-Natal, South Africa
QuSoft, University of Amsterdam, Science Park 123, 1098 XG Amsterdam, The Netherlands

**Robbie King**
Department of Computing and Mathematical Sciences, Caltech, Pasadena, CA 91125, USA

**Hsin-Yuan Huang**
Department of Computing and Mathematical Sciences, Caltech, Pasadena, CA 91125, USA
Institute for Quantum Information and Matter, Caltech, Pasadena, CA 91125, USA

**William J. Huggins**
Google Quantum AI, Venice, California 90291, USA

**Ramis Movassagh**
Google Quantum AI, Venice, California 90291, USA

**Dar Gilboa**
Google Quantum AI, Venice, California 90291, USA

**Jarrod R. McClean**
Google Quantum AI, Venice, California 90291, USA
`jmcclean@google.com`

## Abstract

The success of modern deep learning hinges on the ability to train neural networks at scale. Through clever reuse of intermediate information, backpropagation facilitates training through gradient computation at a total cost roughly proportional to running the function, rather than incurring an additional factor proportional to the number of parameters – which can now be in the trillions. Naively, one expects that quantum measurement collapse entirely rules out the reuse of quantum information as in backpropagation. But recent developments in shadow tomography, which assumes access to multiple copies of a quantum state, have challenged that notion. Here, we investigate whether parameterized quantum models can train as efficiently as classical neural networks. We show that achieving backpropagation scaling is impossible without access to multiple copies of a state. With this added ability, we introduce an algorithm with foundations in shadow tomography that matches backpropagation scaling in quantum resources while reducing classical auxiliary computational costs to open problems in shadow tomography. These results highlight the nuance of reusing quantum information for practical purposes and clarify the unique difficulties in training large quantum models, which could alter the course of quantum machine learning.

37th Conference on Neural Information Processing Systems (NeurIPS 2023).

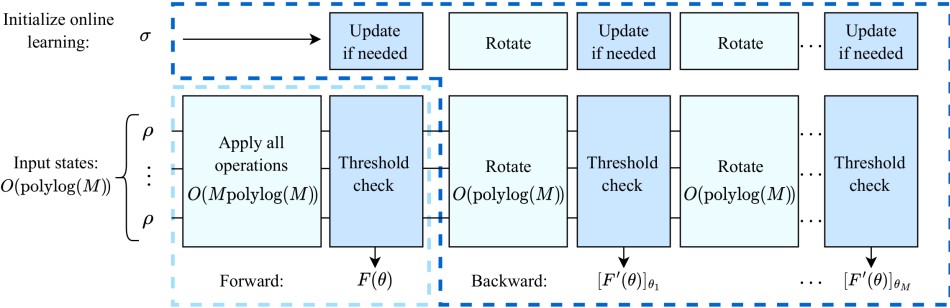

Figure 1: **Quantum backpropagation algorithm.** Our proposal for quantum backpropagation consists of an online shadow tomography protocol, coupled with a threshold search procedure [Aaronson et al., 2018, Bǎdescu and O'Donnell, 2021]. The algorithm is executed in batches of size $O(\text{polylog}(M))$, of which roughly $n$ batches are needed, where $\rho$ is an $n$ qubit quantum state. A classically constructed hypothesis state $\sigma$ is also necessary for the algorithm. Crucially, quantum states *and* the hypothesis state are rotated before each threshold check, to rotate through the layers of a quantum neural network $F(\theta)$, $\theta \in \mathbb{R}^M$ and reuse information for gradients. This enables a cost reduction from $O(M^2 \cdot \text{polylog} M)$ to $O(M \cdot \text{polylog}(M))$ to compute the full gradient. For convenience, we suppress precision factors which scale as $O(1/\varepsilon^4)$ for this proposal.

# 1 Introduction

Computing gradients through backpropagation is crucial to the success of modern deep neural networks. Rather than a naive manifestation of the chain rule to compute gradients, backpropagation leverages white-box knowledge of a computational graph, as well as intermediate values, to asymptotically improve run times [Goodfellow et al., 2016, Rumelhart et al., 1986, LeCun et al., 2015, 1989, Bengio et al., 2014]. Remarkably, computing the gradient of, say, a neural network function with respect to all its parameters, can be done at a total cost roughly proportional to running the function, instead of incurring an additional factor proportional to the number of parameters. This relative scaling, owed to backpropagation, has facilitated the training of very deep networks, with parameter counts now in order of $10^{10}$ – accompanied with unparalleled empirical success [Goodfellow et al., 2016, Szegedy et al., 2017, Iandola et al., 2016, Wu et al., 2019]. When considering the number of function calls required to compute gradients, backpropagation in classical circuits remains exponentially more efficient with respect to the number of parameters, than the best known algorithms for determining gradients of parameterized quantum circuits – with or without the aid of a quantum computer [Gilyén et al., 2019, Van Apeldoorn et al., 2020, Brandão et al., 2017, Jordan, 2005, Schuld and Killoran, 2022]. Nevertheless, the allure of large-scale models inspires the need for efficient training of parameterized quantum models [Kandala et al., 2017, Farhi et al., 2014, Cerezo et al., 2021], which frequently arise in fields like quantum machine learning and quantum chemistry. But if backpropagation scaling cannot be matched, practically reaching overparameterized regimes may be impossible, even before accounting for additional challenges like barren plateaus [McClean et al., 2018, Wang et al., 2021, Cerezo et al., 2020]. Since trainability influences the power and applicability of a model, this could radically shift the current trajectory of preferred quantum models.

In this work, we provide an operational definition of backpropagation, and subsequently determine its feasibility for parameterized quantum models. We investigate learning algorithms with and without quantum memory, where the former is able to store a product of multiple copies of a particular state, perform joint quantum operations followed by an entangled measurement. Whereas, a learning algorithm without quantum memory can only perform operations on each copy, implement a (conditional) measurement, and use the resulting classical data. Without access to multiple copies, we highlight that all known methods to compute gradients of simple variational models, do not achieve an overhead in line with backpropagation, unless one considers very special cases. Interestingly, closely related probabilistic classical analogues can exhibit backpropagation scaling, which points out that the barrier in the quantum setting is due to quantum phenomena. In an attempt to mimic classical backpropagation, which leverages information reuse to produce a favourable scaling, we lean on a similar concept in a quantum setting, namely *gentle measurements* [Aaronson, 2019, Aaronson and Rothblum, 2019, Bǎdescu and O'Donnell, 2021]. When combined with online learning [Aaronson

et al., 2018], this technique has proven useful in problems like *shadow tomography* as it aims to conserve quantum resources, but has yet to be explored in the context of backpropagation. In an information-theoretic sense, when access to multiple copies of a state is provided, a modification of existing shadow tomography routines enables backpropagation scaling if one restricts costs to the quantum overhead and ignores the classical cost incurred to implement known shadow tomography schemes. We present our proposed quantum backpropagation algorithm in Figure 1 which highlights the reduction in quantum resources due to the ability to exploit structure in a quantum neural network and reuse information through gentle measurement. Unfortunately, the true computational efficiency of our scheme remains an open question and we rule out a general strategy based on gentle measurement alone, by linking to computational models known to be more powerful than those contained in the complexity class of BQP [Aaronson et al., 2016]. Despite failure to achieve backpropagation scaling in this general setting, the construction is suggestive of approximate or restricted models that may yield the desired scaling without violating complexity-theoretic bounds. This avenue remains rich for future work. We hope that these results illustrate the difficulty of replicating backpropagation scaling in parameterized quantum circuits and inspires the development of alternative quantum models that can train at scale.

## 2  Backpropagation scaling

A comprehensive overview of gradient evaluation, given by automatic differentiation on classical computers, can be found in Griewank and Walther [2008]. The key advantage, however, can be summarized in one sentence: computational and memory resources employed to compute gradients of a function are bounded multiples of those used to compute the function. We use this bound to define the requirements for backpropagation scaling.

**Definition 1** (Backpropagation scaling). Given a parameterized function $F(\theta)$, $\theta \in \mathbb{R}^M$, let $F'(\theta)$ be an estimate of the gradient vector accurate to within some constant $\varepsilon$ in the infinity norm. The total computational cost incurred to obtain $F'(\theta)$ with backpropagation is bounded such that

$$\text{TIME}(F'(\theta)) \leq c_t \cdot \text{TIME}(F(\theta)), \tag{1}$$

and

$$\text{MEMORY}(F'(\theta)) \leq c_m \cdot \text{MEMORY}(F(\theta)), \tag{2}$$

where $c_t, c_m = \text{polylog}(M)$, and $\text{TIME}(\cdot)$ and $\text{MEMORY}(\cdot)$ capture the time and space complexity respectively, for either computing the function $F$ or its gradient $F'$.

As a further specification of backpropagation scaling in Definition 1, one can specify whether one achieves this scaling in quantum resources, classical resources, or all resources. While it is, of course, the goal to achieve this scaling in all resources, the distinction remains relevant due to the ability to leverage classical resources in order to improve the scaling in quantum resources, which we elaborate on in Section 5. In purely classical models, like neural networks, the overhead for both time and memory can be constant, and typically by a small factor. This efficiency has been instrumental for training very large models and is arguably the main contributor to the success of modern day machine learning. Given that variational quantum models, which utilize parameterized quantum circuits, are believed to be the most promising candidates to solve quantum machine learning tasks, we investigate their ability to reproduce this scaling.

## 3  Variational quantum models

Variational algorithms have become a go-to approach when looking to solve various optimization and machine learning problems on quantum devices [Cerezo et al., 2021]. We present a slightly restricted model for ease of analysis which still covers a very broad range of practical scenarios. Notably, if backpropagation scaling cannot be achieved in this simplified setting, it is unlikely to succeed in a more sophisticated one.

**Definition 2** (Simple variational model). Consider an initial quantum state $\rho$ and a quantum circuit with $M$ parameterized operations $U_j(\theta_j) = e^{-i\theta_j P_j}$, where each $P_j$ is a Pauli operator acting on up to $n$ qubits. We define a simple variational quantum model as the parameterized function

$$F(\theta) = \text{Tr}[O\rho(\theta)], \tag{3}$$

where $O$ is a Hermitian and unitary observable, and the quantum state $\rho(\theta)$ is expressed as $\rho(\theta) = U(\theta)\rho U(\theta)^\dagger$. In the most general case, $\rho$ will be an unknown quantum state that we refer to as the quantum data setting, but we will also be interested in the simplified setting where $\rho(\theta) = |\psi(\theta)\rangle\langle\psi(\theta)|$ and $|\psi(\theta)\rangle = U(\theta)|0\rangle^{\otimes n} = \prod_{j=1}^M U_j(\theta_j)|0\rangle^{\otimes n}$.

In the simplified setting, the $k^{\text{th}}$ gradient component of $F(\theta)$ can be expressed as

$$[F'(\theta)]_{\theta_k} = 2\,\mathrm{Re}\left[\langle 0|\left(\prod_{j=1}^M e^{i\theta_j P_j}\right) O \left(\prod_{m=k+1}^M e^{-i\theta_m P_m}\right)(-iP_k)\left(\prod_{l=1}^k e^{-i\theta_l P_l}\right)|0\rangle\right]. \quad (4)$$

It becomes clear that computing all $M$ components involves a large number of common operations. At face value, one might think it straightforward to exploit this overlap of operations to gain computational efficiency, as is done classically. The problem is that intermediate information in a quantum circuit is not easily retrievable without consequence, which is investigated in the following section.

## 4  Learning algorithms without quantum memory

Recall that a learning algorithm without quantum memory would perform operations and measurements on each individual copy of a quantum state. In this regime, which is prominent in current quantum machine learning settings, we have the following proposition.

**Proposition 3** (Backpropagation scaling is impossible for quantum data using single copies). *Given the quantum data setting where one seeks to train a variational model using copies of the unknown state $\rho$ and the additional constraint of no quantum memory, then backpropagation scaling is not possible in the general case.*

*Proof.* Take the Pauli circuit model above and let us consider the case of all possible Pauli operators $P_j$ on $n$ qubits, such that $M = O(4^n)$. If we take the special case of quantum data and initializing all $\theta_j = 0$, then the gradient with respect to each of the parameters is given by the expected value of all possible Pauli operators on $n$ qubits on the unknown quantum state $\rho$, up to a small constant. If no quantum memory is available, that is, we only have the ability to perform measurements on single copies at a time, then by Chen et al. [2022, Corollary 5.9], the minimal number of copies of $\rho$ is lower bounded by $\Omega(2^n/\varepsilon^2)$ in order to predict all Pauli operators to at most $\varepsilon$-error with probability $2/3$. Hence, backpropagation scaling is not possible in general in the single copy case. $\qquad\square$

Notably, Proposition 3 is based on an information-theoretic separation that does not generalize to the simplified case, $\rho = |0\rangle\langle 0|$, or even when $\rho$ is simply guaranteed to be a pure state generated by a polynomial sized circuit, which we detail in Appendix C. Hence, for the simplified case and polynomial complexity pure state cases, we must turn to computational arguments. Furthermore, if it were possible to find a polynomial time algorithm for the approach in Appendix C, then it would be possible to efficiently clone pseudo-random states, which is not believed to be possible [Ji et al., 2018], despite the fact that they are pure states generated by polynomial sized circuits (see Appendix C.2). The following remark aims to clarify the status of current methods for approaching this problem.

**Remark 4** (Current gradient methods fail to achieve backpropagation scaling). Given a variational model $F(\theta)$ defined in (3) with time complexity

$$\mathrm{TIME}(F(\theta)) = \tilde{O}(M/\varepsilon^k),$$

for some integer $k$ and precision $\varepsilon$, then all known schemes to estimate the gradient of $F(\theta)$ to the same precision, do not, in general, achieve a time complexity in line with backpropagation scaling.

We briefly explain why known gradient methods fail, but defer details to Appendix A. A promising gradient algorithm put forth in Jordan [2005] requires only a single black-box query to a function to estimate its full gradient with a desired precision. But, as shown in Gilyén et al. [2019], when considering variational models, a different query model must be applied and the original single-query advantage becomes unattainable. The authors derive lower bounds requiring a quantum computational cost of $O(M\sqrt{M}/\varepsilon^2)$ and in a high precision regime, $O(M\sqrt{M}/\varepsilon)$ is worst-case

optimal [Huggins et al., 2021] when using a black-box simplified model of $U(\theta)$. In other contexts, it is also sometimes argued that the simultaneous perturbation stochastic approximation (SPSA) algorithm is computationally efficient since it requires two function evaluations to estimate the gradient, irrespective of $M$. This seemingly satisfies the scaling required, however, as $M$ increases, the variance of the gradient estimate increases and, thus, to counteract this, either a smaller learning rate must be used - increasing the number of optimization steps - or more samples are needed to estimate the gradient with an appropriate accuracy at every step. We derive a sample complexity bound in Appendix A.2.3 which demonstrates SPSA's inability to exhibit backpropagation scaling. Thus far, other sampling schemes constructed to estimate the gradient of $F(\theta)$, like the parameter-shift rule, perform destructive measurements that typically only retrieve a partial amount of information for one component of the gradient. As a result, reducing the infinity norm error in the gradient with reasonable probability, has a cost that scales like converging each component, i.e.

$$\text{TIME}(F'(\theta)) \propto M \ \text{TIME}(F(\theta)) \tag{5}$$

$$= \tilde{O}(M^2/\varepsilon^k), \tag{6}$$

which unfortunately, does not achieve backpropagation scaling.

While this quadratic dependence on the number of parameters may not seem problematic, a linear dependence was the necessary catalyst in the age of modern deep learning, with overparameterized networks that perform exceedingly well on practical tasks. We illustrate the consequences of quadratic scaling in Appendix A.2, Figure 2, where one could wait up to a day to evaluate a single gradient estimate of a model with fewer than 10 000 parameters.

But perhaps neural networks are not a fair benchmark. One could dig deeper in automatic differentiation literature to investigate whether a direct classical analogue for these parameterized quantum circuits attains backpropagation scaling. Interestingly, a particular analogue can.

**Proposition 5** (Classical analogue achieves backpropagation scaling). *Parameterized Markov chains, which are much closer classical analogues to variational models than neural networks, exhibit backpropagation scaling.*

We detail the proof and scaling comparison in Appendix B by drawing an analogy between quantum and classical probabilistic states. Under some reasonable assumptions on the set of classical operations, the desired scaling is indeed possible in analogous classical parameterized stochastic processes. The formulation of this classical-quantum analogy allows us to probe the root cause of why backpropagation scaling is so difficult to obtain in the quantum variational setting. The origin of the challenge lies within quantum measurement collapse and the inability to read out intermediate states while continuing a computation, rather than the probabilistic formulation of the problem. In the classical setting, one is always promised to be in a computational basis state, making it possible to do perfect measurements non-destructively at intermediate steps. It remains an interesting open question to better understand the performance separation on practical tasks between quantum variational methods and this type of classical analogue, given the advantage in trainability of the latter.

Although Proposition 3 presents a strict lower bound ruling out backpropagation in the quantum data case with single copies, this leads one to wonder whether backpropagation scaling is possible when one has access to multiple copies. Moreover, destructive quantum measurements are the inhibitor of backpropagation scaling in single copies, so perhaps there is some middle ground where one could perform measurements that are only partially destructive on multiple copies. This idea has led to breakthroughs in the shadow tomography problem, which we examine next in the context of backpropagation.

## 5  Reusing multiple copies through gentle measurement

By allowing access to multiple copies of $\rho$, it is especially interesting to note that gentle measurements can facilitate backpropagation scaling in all resources, when considering the special case outlined in Proposition 3. We first define gentle measurement, followed by the special case construction.

**Definition 6** (Gentle measurement). Fix a subset of quantum mixed states $\mathcal{S}$. A measurement $F$ is *$\alpha$-gentle on $\mathcal{S}$* if for every state $\rho \in \mathcal{S}$, and every outcome $y$ of $F$, the post-selected state $\rho_{F=y}$ obeys

$$||\rho_{F=y} - \rho|| \leq \alpha,$$

where $\alpha \in [0, 1]$. Hence, the smaller the $\alpha$, the less damage incurred by $\rho$.

**Proposition 7** (A special case variational model achieves backpropagation scaling). *Given a variational model $F(\theta) = \mathrm{tr}\big[U(\theta)\rho U(\theta)^\dagger\big]$, where $U(\theta) = \prod_{j=1}^M e^{-i\theta_j P_j} V$ for some fixed unitary $V$, setting $\theta$ to zero and $O = I$, the $k^{th}$ gradient component may be written as*

$$F'(0)_k = 2\,\mathrm{Re}\left(\mathrm{Tr}\big[V\rho V^\dagger(-i)P_k\big]\right) = 2\,\mathrm{Im}\left(\mathrm{Tr}\big[V\rho V^\dagger P_k\big]\right). \tag{7}$$

*Then all $M$ gradient components can be estimated to within a fixed precision $\varepsilon$ using $O(\log(M)/\varepsilon^4)$ function calls, and thus,*

$$\mathrm{TIME}(F'(\theta)) = O(\log(M))\mathrm{TIME}(F(\theta)),$$

*which is in line with backpropagation scaling.*

*Proof.* With use of an ancilla qubit and a slightly modified circuit, the imaginary components of $\mathrm{Tr}\big[V\rho V^\dagger P_k\big]$, which represent gradient components, can be estimated with $\tilde{O}(\log M/\varepsilon^4)$ copies of $\rho$ for all $k$ via application of V and two-copy Bell measurements of the resulting state that harness gentleness. Similarly, using $O(\log(M)/\varepsilon^2)$ copies along with the magnitude information from the Bell measurements, one may estimate the sign of the gradient components using a majority vote scheme that also exploits gentleness. The total number of copies scales as $O(\log(M)/\varepsilon^4)$, inducing a time complexity of $O(\log(M) \cdot M/\varepsilon^4)$ with efficient classical overhead. The details of the implementation are discussed in Huang et al. [2021, Appendix E]. $\square$

This result indicates that there are at least some choices of circuits and generators for which backpropagation scaling can be achieved using gentle measurements. The exception naturally leads one to ask if this may be possible in more general cases with techniques like shadow tomography [Aaronson, 2019], however, with just a small perturbation away from this special case, the same technique no longer works, and the general computational efficiency remains unknown [Aaronson and Rothblum, 2019, Bǎdescu and O'Donnell, 2021]. In the subsequent section, we adapt shadow tomography results and exploit the sequential structure in variational models equipped with quantum data, to obtain backpropagation scaling in quantum resources, but leave open the question of classical computational efficiency. This represents substantial progress over current gradient methods for these models.

### 5.1 A quantum-efficient protocol for backpropagation

Our main contribution in this more general quantum data setting with multi-copy access, is the establishment of a connection between gradient estimation and shadow tomography. This gives an exponential improvement to the sample complexity of the input state from $\tilde{O}(M)$ to $O(\mathrm{polylog}\,(M))$ for computing gradients. It also gives a quadratic improvement in the number of quantum operations from $\tilde{O}(M^2)$ to $\tilde{O}(M)$, analogous to classical backpropagation. The algorithm is depicted diagrammatically in Figure 1. Our proposal, however, houses a large caveat: it requires the classical storage and manipulation of a *hypothesis state*, which results in an exponential classical overhead, unless an approximation scheme can be effectively applied. It is argued in Aaronson [2019] that this cost is unavoidable in general, since removing it would imply that quantum advice can always be simulated by classical advice. Nevertheless, the exponential saving in sample complexity could be important in settings where the labelled quantum states coming from Nature are limited, and valuable. In Huang et al. [2022] for example, there were sources of quantum data that, when limited in quantity, could achieve a substantial data advantage over classical learners – even in the range of 20-40 qubits. In this size range, keeping the classical model in full detail would be completely feasible without ruining the potential for quantum advantage. Further, the linear scaling of quantum operations, even in the face of exponential classical overhead, could be beneficial if classical computation is extremely cheap when compared to quantum computation.

Our protocol will apply to an even more general model than Equation (3), which we term a quantum neural network.

**Definition 8** (Quantum neural network). Let a quantum neural network be a variational quantum circuit on $n + 1$ qubits, numbered $0, 1, \ldots, n$. Qubits $1, \ldots, n$ act as the data register, which will take as input an unknown quantum state $|\varphi\rangle$ to be classified. Qubit 0 acts as the output register, which is measured in the $Z$-basis and initialized to $|0\rangle$. The variational circuit belongs to the following simple class

$$\mathcal{U}(\vec{\theta}) = e^{i\theta_M P_M} U_M \ldots e^{i\theta_1 P_1} U_1,$$

where $\{P_k\}$ are fixed $(n+1)$-qubit Pauli operators and $\{U_k\}$ are fixed circuits. The output prediction on $|\varphi\rangle$ is then given by a quantum neural network function defined as

$$\text{QNN}_{\vec{\theta}}(|\varphi\rangle) = \langle 0|\langle\varphi|\mathcal{U}^\dagger(\vec{\theta})\, Z_0\, \mathcal{U}(\vec{\theta})|0\rangle|\varphi\rangle \in [-1, 1]. \tag{8}$$

Note that running the circuit on $|0\rangle|\varphi\rangle$ gives a coin flip $\text{Ber}(\frac{1}{2} + \frac{1}{2}\text{QNN}_{\vec{\theta}}(|\varphi\rangle))$ rather than $\text{QNN}_{\vec{\theta}}(|\varphi\rangle)$ itself. This allows us to estimate $\text{QNN}_{\vec{\theta}}(|\varphi\rangle)$ to $\varepsilon$ precision with high probability by running the circuit $\text{poly}(\varepsilon^{-1})$ times, as usual. Furthermore, note the sequential nature of the function's gradients, highlighted in a similar sense in Equation (4). This leads us to the following theorem.

**Theorem 9** (Quantum-efficient backpropagation). *Given an unknown $n$ qubit input state $|\varphi\rangle$, there exists an explicit algorithm which produces estimates $b_k$ for all $k = 1, ..., M$ such that $|b_k - \frac{1}{2}\partial_{\theta_k}QNN_{\vec{\theta}}(|\varphi\rangle)| \leq \varepsilon$ using only*

$$m = O\left(\frac{n\log^2 M}{\varepsilon^4}\right),$$

*copies of $|\varphi\rangle$. The required number of quantum operations for the proposed algorithm is $\tilde{O}(mM)$, which is quasi-linear in $M$. However, classical storage of a hypothesis state is used and incurs a classical cost of $M \cdot 2^{\tilde{O}(n)}$ when no effective approximation schemes are known.*

The full details of the proof and the explicit quantum backpropagation algorithm are given in Appendix D. We first show that estimating the gradient component $\partial_{\theta_k}\text{QNN}_{\vec{\theta}}(|\varphi\rangle)$ reduces to estimating the expectation value of a certain traceless Hermitian unitary operator on $|+\rangle|0\rangle|\varphi\rangle$. Shadow tomography results then imply that estimating all gradient components to precision $\varepsilon$ is possible using only $\text{poly}(\log M, n, \varepsilon^{-1})$ copies of $|\varphi\rangle$. In order to fully specify the algorithm, we adapt an improved shadow tomography protocol from Bădescu and O'Donnell [2021] that makes use of gentle measurements and is online. The key difference in our proposal, which enables us to achieve linear scaling in $M$, is the reuse of quantum computation in a way reminiscent of backpropagation through observation that one can rotate through the layers of the quantum neural network sequentially and estimate the appropriate expectation value between each rotation step, as shown in Figure 1. Naive implementation of the shadow tomography protocol for gradients would yield $\tilde{O}(M^2)$ quantum operations, in line with most existing methods for quantum gradient estimation.

## 5.2 Reduction from shadow tomography

We now show that a fully efficient algorithm for computing gradients would give rise to a fully efficient shadow tomography procedure for observables which can be efficiently implemented. This very general class of observables, however, is not known to have a computationally efficient shadow tomography protocol. Thus, this connection presents yet another obstacle to improving the exponential classical run time of our quantum backpropagation algorithm since removing the $\exp(n)$ classical run time overhead in general, necessitates a breakthrough in shadow tomography.

**Definition 10** (Shadow tomography problem). Let $\mathcal{E}$ be a class of two-outcome measurements with outcomes in $\{\pm 1\}$. Given an unknown $n$-qubit quantum state $|\psi\rangle$, and known measurements $E_1, \ldots, E_M \in \mathcal{E}$, output estimates $b_1, \ldots, b_M \in [-1, 1]$ such that $|b_k - \langle\psi|E_k|\psi\rangle| \leq \varepsilon\, \forall k$. In particular, do this via a measurement of $|\psi\rangle^{\otimes m}$ where $m$ is as small as possible.

**Definition 11** (Poly-time observables). A *poly-time observable* on $n$ qubits is defined to be an observable of the form $U^\dagger Z_1 U$ where $U$ is a poly-size circuit.

The shadow tomography problem is well-studied in quantum information theory. There are indeed special cases where this problem may produce a favourable scaling in $M$ and $n$, as outlined in Proposition 7. But, in general, it is not trivial to remove the exponential classical cost when it comes to shadow tomography.

**Theorem 12** (Shadow tomography reduction). *Suppose there is an algorithm which can estimate the gradients $\partial_{\theta_k}QNN_{\vec{\theta}}(|\psi\rangle)$, $k = 1, \ldots, M$, to precision $\varepsilon$, with $m$ copies of $|\psi\rangle$, and with runtime $T$. Then, this gives an algorithm for shadow tomography of poly-time observables, to precision $\frac{\varepsilon}{2}$, with $m$ copies of $|\psi\rangle$ and runtime $T$.*

*Proof.* Consider an instance of shadow tomography on $n$ qubits, with $E_1, \ldots, E_M$ given by $E_k = U_k^\dagger Z_1 U_k$, where $\{U_k\}$ are poly-size circuits. Construct the quantum neural network with the following

variational circuit

$$\mathcal{U}(\vec{\theta})|0\rangle|\psi\rangle = e^{i\theta_M Y_0 \otimes Z_1}\hat{U}_M \ldots e^{i\theta_1 Y_0 \otimes Z_1}\hat{U}_1 H_0 |0\rangle|\psi\rangle$$
$$= e^{i\theta_M Y_0 \otimes Z_1}\hat{U}_M \ldots e^{i\theta_1 Y_0 \otimes Z_1}\hat{U}_1 |+\rangle|\psi\rangle$$

where

$$\hat{U}_1 = \mathbb{1}_0 \otimes U_1$$
$$\hat{U}_k = \mathbb{1}_0 \otimes U_k U_{k-1}^\dagger , \quad 1 < k \le M$$

Then, by Equation (7), the gradients at $\vec{\theta} = \vec{0}$ are

$$\partial_{\theta_k}\mathrm{QNN}_{\vec{\theta}}(|\psi\rangle)|_{\vec{\theta}=\vec{0}} = 2\,\mathrm{Re}\,\langle 0|\langle\psi|\mathcal{U}^\dagger(\vec{0})Z_0\partial_{\theta_k}\mathcal{U}(\vec{0})|0\rangle|\psi\rangle$$
$$= 2\,\mathrm{Re}\,\langle +|\langle\psi|\hat{U}_1^\dagger \ldots \hat{U}_M^\dagger Z_0 \hat{U}_M \ldots \hat{U}_{k+1}(iY_0 \otimes Z_1)\hat{U}_k \ldots \hat{U}_1|+\rangle|\psi\rangle$$
$$= 2\,\mathrm{Re}\,\langle +|\langle\psi|\hat{U}_1^\dagger \ldots \hat{U}_k^\dagger (iZ_0(Y_0 \otimes Z_1))\hat{U}_k \ldots \hat{U}_1|+\rangle|\psi\rangle$$
$$= 2\langle +|\langle\psi|(\mathbb{1}_0 \otimes U_k^\dagger)(X_0 \otimes Z_1)(\mathbb{1}_0 \otimes U_k)|+\rangle|\psi\rangle$$
$$= 2\langle +|\langle\psi|(X_0 \otimes E_k)|+\rangle|\psi\rangle$$
$$= 2\langle\psi|E_k|\psi\rangle$$

Thus computing the gradients allows us to solve the shadow tomography instance. $\qquad\square$

Seeing as there is no known classically efficient procedure for shadow tomography with respect to poly-time observables, this reduction illustrates the difficulty of replicating true backpropagation scaling in general.

### 5.3 A fully gentle gradient strategy

Shadow tomography makes use of multiple copies and a hypothesis state model, often stored classically, to require a minimal number of destructive measurements. It is useful to examine the limits of gentle measurement alone for gradient estimation in order to reduce the classical overhead. In particular, it would be ideal if it were possible to use a small number of copies (e.g. $\mathrm{polylog}(1/\alpha)$) of a quantum state to achieve $\alpha$-gentleness in the general case through a simple, direct measurement scheme. While we do not explicitly construct a protocol here, this capability would naturally lead to a scheme for gradient estimation that achieves backpropagation scaling. This capability, however, would also allow us to violate known query lower bounds for the unstructured search problem. Thus, for our gradient purposes, it seems as if successful schemes must limit the number of potentially destructive accesses to a quantum state via the use of a classical model. We formalize the general failure of gentle measurement alone in the following theorem.

**Theorem 13** (Repeated gentle measurements). *Assume it is possible to perform an arbitrary two-outcome measurement gently by using up to a polylogarithmic number of copies of the state. Specifically, assume that any measurement can be made $\alpha$-gentle by using $O(\mathrm{polylog}(1/\alpha))$ copies of the state. Such an ability leads to a violation of known query bounds given by Grover's search algorithm, and thus, cannot be possible in general.*

*Proof.* The proof is adapted from results in Aaronson et al. [2016]. Consider the $n$-qubit Grover state after $i$ iterations with an ancilla present to mark the state $|x\rangle$,

$$\sin((2i+1)\theta)|x\rangle|1\rangle + \cos((2i+1)\theta)\sum_{y\in\{0,1\}^n\ y\ne x}\frac{1}{\sqrt{M-1}}|y\rangle|0\rangle, \qquad (9)$$

where $\theta = \arcsin 2^{-\frac{M}{2}}$. For each of the $M = 2^n$ possible marked elements $x$, one can define a two-element POVM of the form $\{|x\rangle|1\rangle\langle x|\langle 1|,\ I - |x\rangle|1\rangle\langle x|\langle 1|\}$. One may ensure that the marked bit string is found with high probability by performing a measurement of each of these POVMs with respect to the Grover state $\tilde{O}(2^n)$ times, even in the case where the state is constructed with a single Grover oracle query. Performing this procedure using standard, destructive, measurements of the POVMs would require a fresh set of oracle queries with each round. However, using sufficiently gentle measurements removes this requirement. If the distance between the pre- and post-measurement

states is sufficiently small, one obtains results that are close to those that one would obtain from a fresh copy of the state. In order to guarantee that the marked bit string can be extracted with high probability, we demand that the state obtained after any number of measurement rounds be within $\frac{2^{-n}}{3}$ in the trace distance of the state prior to any measurements[1]. To guarantee that the state be sufficiently unchanged by the end of the series of $\tilde{O}(2^{2n})$ measurements, each measurement should therefore be $\tilde{O}(2^{-3n})$-gentle. By assumption, this is possible with $\mathrm{polylog}(2^{3n})$, or simply $\mathrm{poly}(n)$, copies of the original state, each of which is prepared using the Grover oracle a set number of times. By performing the whole sequence of measurements gently, one can avoid biasing the result too much before the marked state is found. Hence, one can identify $x$ with high probability, using $\mathrm{poly}(n) \ll 2^{n/2}$ Grover oracle calls, which is a violation of known lower bounds [Grover, 1996]. In Appendix E.4, we discuss how sufficiently gentle measurements lead to a violation of known bounds when considering a different notion of time complexity that combines measurements and oracle queries. □

**Remark 14** (Shadow tomography does not violate known bounds). After seeing this result, one might question how this relates to shadow tomography schemes that use gentle measurement plus classical computation. It is consistent when one considers that $\alpha$-gentleness considered in isolation must apply to both the number of distinct measurements one may perform and the precision to which one performs a particular repeated measurement. That is, from the point of view of gentle measurement alone, gentleness on different measurements and gentleness on repeated measurement to high precision $\varepsilon$, are on the same footing and hence, must respect known bounds for information extraction. Indeed, all known shadow tomography schemes are consistent with a number of copies of the state scaling polynomially with $1/\varepsilon$, despite scaling logarithmically in the number of distinct measurements, which prevents the above violation of known Grover query and time complexity bounds. This reflects an asymmetry between the number of distinct measurements and the precision of a single measurement present in all shadow tomography schemes and noted in the original work on the topic [Aaronson, 2019] that hypothesized that there are fewer independent observables within a quantum state than one might expect intuitively. The success of shadow tomography schemes, as distinct from simple gentle measurement, depends crucially on the existence of models that update quickly enough to limit the number of measurements made to the actual quantum states.

## 5.4 Approximate schemes

The failure of a fully gentle approach points to the necessity of a classical model to enable backpropagation scaling. But, the key challenge in the general application of the proposed shadow tomography algorithm is the use of an explicit classical representation of the quantum state, which in general, scales exponentially with system size. While there have been a few special cases found that have fully efficient schemes, like with Pauli operators [Huang et al., 2021], the case of whether there exists an efficient computational scheme for poly-time observables remains open. However, an exact scheme may not be required in practice, especially when dealing with noisy data. This raises the possibility of using approximate classical representations of the state. For example, it is known that in cases where states exhibit low entanglement, they may be efficiently represented by matrix product or tensor network states [White, 1992, Perez-Garcia et al., 2006, Cramer et al., 2010, Evenbly and Vidal, 2015, Orús, 2019]. Moreover, in the case of shadow tomography, one is not explicitly seeking an exact representation of the density matrix, but rather a proxy, capable of reproducing the desired observables with high probability. This relaxation of requirements may render an approximation scheme effective, even when the true state is challenging to represent with a particular ansatz. This area represents an interesting and potentially fruitful research direction that could dramatically increase the efficiency of training in quantum machine learning models, and we leave it open for future work.

## 6 Discussion

Special cases aside, the inadequacy of current gradient methods to provide backpropagation scaling in parameterized quantum models leaves room for many questions. One particular conclusive direction

---

[1]This choice of distance guarantees that, even in the presence of damage, the POVM used to identify the actual marked element will correctly return a positive result with probability at least $\frac{2}{3}2^{-n}$. Likewise, any POVM corresponding to an unmarked element will incorrectly return a positive result with probability at most $\frac{1}{3}2^{-n}$.

would be developing a concrete computational argument to rule out backpropagation scaling in the multi-copy setting, thereby confirming the true computational complexity of shadow tomography. Even though the proposed information-efficient scheme in this study fails to satisfy classical cost requirements of backpropagation, the possibility of a computationally efficient procedure remains open, especially for cases with known, structured observables. Similarly, failure in the general case of gentle measurements again suggests potential approximate or restricted models that may be more trainable. One may find an alternative architecture where gradient computation has favorable scaling, and even though the model is perhaps not as powerful or universal, it may still be useful in practice. Interestingly, closely related probabilistic classical analogues to variational models can exhibit backpropagation scaling. If the difficulty to achieve an efficient scaling is due to inherently quantum properties, perhaps backpropagation is not the correct method for optimization of quantum models, which seems to be a growing belief for classical models too, albeit for completely different reasons [Hinton, 2022]. We hope that these results spark the development of either alternative quantum models that can train at scale or new methods for efficient optimization.

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

# A Resource scaling for quantum backpropagation methods

What comprises classical memory and time complexity, is purposely left vague. The details depend on the constituent types of operations needed to compute a function and its gradients, as well as the memory access model available. But, details aside, backpropagation merely refers to gradient computation in a particular manner, and, any reasonably successful implementation of it incurs a constant overhead in relative complexity, as captured by Equations (1) and (2). With this in mind, we elaborate on the operational definition of quantum backpropagation scaling in terms of memory. Thereafter, we explain the failure of various current gradient methods to achieve backpropagation scaling.

## A.1 Memory complexity of the function

Recall the function of interest $F(\theta) = F(\theta) = \text{tr}[\rho(\theta)O]$, where $O$ is an observable and $\rho(\theta)$ is a parameterized quantum state built from $M$ parameters, acting either on an unknown initial state $\rho$ or simplified initial state $\rho = |0\rangle\langle 0|$. Classifying the memory used to compute the function as a combination of $n$ qubits, plus storage for each of the $M$ parameters with appropriate precision, $\delta$, implies

$$\text{MEMORY}(F(\theta)) = \tilde{O}(n + M\log(1/\delta)). \tag{10}$$

To derive the computational cost, assume unit cost access to any element of the circuit family $\{U_j\}$. If an incoherent measurement scheme is used, measuring $O$ and estimating $F(\theta)$ to an acceptable fixed precision, $\varepsilon$, on repeated preparations of $\rho(\theta)$ incurs a cost that scales as $\text{TIME}(F(\theta)) = \tilde{O}(\frac{M}{\varepsilon^k})$, for some integer $k$. This sets the scene for the computational requirements of computing $F'(\theta)$, which should, importantly, be achieved with a modest space overhead to truly replicate backpropagation.

## A.2 Current gradient methods

Replicating classical backpropagation efficiency in a quantum setting requires more effort, which we elaborate on next by discussing how and why current gradient methods fail to achieve this efficiency. For further illustration, Figure 2 provides a hypothetical comparison between the popular gradient method – the parameter-shift rule – and true quantum backpropagation. The plot incorporates assumptions about time to compute native quantum operations taken from Babbush et al. [2021].

### A.2.1 Naive sampling

The gradient of the function $F(\theta)$ expressed in Equation (4) also takes a simpler form using the parameter-shift rule and properties of Pauli generators [Mitarai et al., 2018, Schuld et al., 2019]

$$[F'(\theta)]_{\theta_k} = F\big(\theta + \frac{\pi}{2}\hat{\theta}_k\big), \tag{11}$$

where $\hat{\theta}_k$ is a unit vector along the $k^{\text{th}}$ direction of $\theta$. Thus far, sampling schemes constructed to estimate (11), perform a destructive measurement that typically only retrieves a partial amount of information for one component of the gradient. As a result, reducing the infinity norm error in the gradient such that we expect $||F'(\theta) - \hat{F}'(\theta)||_\infty \leq \varepsilon$ with reasonable probability, has a cost that scales like converging each component, i.e.

$$\text{TIME}(F'(\theta)) \propto M\log M \ \text{TIME}(F(\theta)) \tag{12}$$

$$= \tilde{O}(M^2/\varepsilon^2). \tag{13}$$

While this quadratic dependence on the number of parameters may not seem problematic, a linear dependence was the necessary catalyst in the age of modern deep learning, with overparameterized networks that perform exceedingly well on practical tasks.

### A.2.2 Fast gradient algorithm

A method put forth by Jordan [2005] numerically estimates the gradient of a classical black-box function at a given point, using a quantum computer. The algorithm impressively requires a single black-box query to estimate the full gradient with a desired precision, whilst satisfying the memory

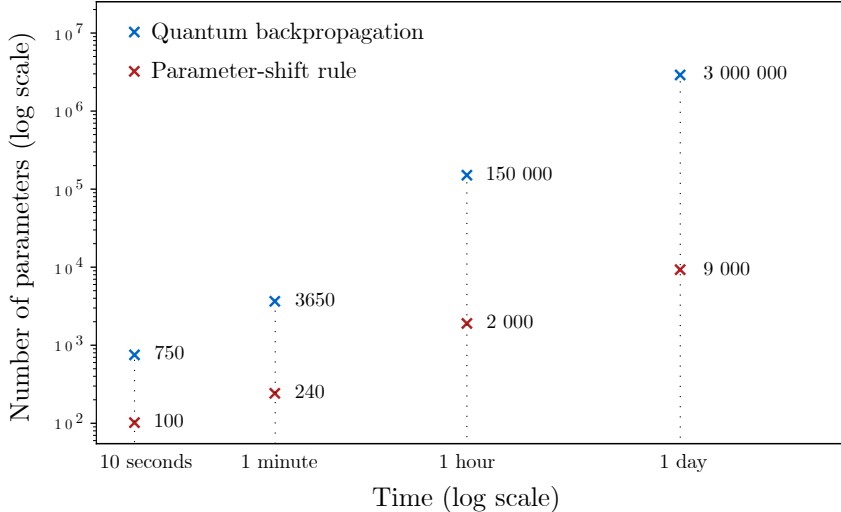

Figure 2: **Quantum backpropagation scaling.** The parameter-shift rule is plotted alongside true quantum backpropagation scaling. On the $x$-axis is time in number of seconds required to compute a single estimate of the gradient in log scale, with common time points stated explicitly. On the $y$-axis is the number of parameters, also in log scale, that may be optimized using each method, for a given amount of time. We make simple assumptions, motivated from the work in Babbush et al. [2021]. Namely, we assume a minimum system size of $n = 100$ qubits. Further, assuming a favourable time of $10\mu s$ to compute one parameterised operation ( which is 1 order of magnitude less than the time to compute one Toffoli gate), the time for one primitive is lower bounded by $100 \times 10\mu s = T_q$. Scaling in time is then roughly $M^2 \cdot T_q$ for the parameter-shift rule and $M \cdot \text{polylog}(M) \cdot T_q$ for quantum backpropagation. Furthermore, $\varepsilon = O(1)$.

requirement in (2). We elaborate on the connection between this approach and backpropagation on a quantum computer when the function considered is classical and reversible, in Appendix B.1. But, as shown by Gilyén et al. [2019], when parameters are considered to be rotation angles like those in variational circuits, a different query model needs to be applied and the original single-query advantage becomes unattainable. With the appropriate query model, the known bounds imply a computational cost of $\tilde{O}(M\sqrt{M}/\varepsilon^2)$ using amplitude estimation, and, in a high precision regime, $\tilde{O}(M\sqrt{M}/\varepsilon)$ is worst-case optimal even with commuting Pauli operators [Huggins et al., 2021]. This worst-case bound was proved in a setting where operators commute, indicating that commutativity need not be helpful in other settings.

### A.2.3    Simultaneous perturbation stochastic approximation (SPSA) algorithm

A few studies have investigated the use of the simultaneous perturbation stochastic approximation (SPSA) algorithm to optimize parameterized quantum circuits [Benedetti et al., 2019, Hoffmann and Brown, 2022, Gacon et al., 2021]. It is argued that SPSA is computationally efficient since its requires two function evaluations to estimate the gradient, irrespective of $M$. This seemingly satisfies the scaling we require, however, the approximation of the gradient has limited accuracy which affects the number of optimization steps needed for SPSA to converge to a minimum. As $M$ increases, the variance of the gradient estimate increases and, thus, to counteract this, a smaller learning rate must be used - increasing the number of optimization steps - or more samples are needed to estimate the gradient with an appropriate accuracy at every step. In either case, one cannot escape a dependence on $M$, which indirectly affects the number of function evaluations needed to estimate gradients or perform gradient-based optimization adequately. More formally, the gradient estimator for component $j$ of a function, given by SPSA, is

$$\bar{F}'(\theta)_j = \frac{F(\theta + c\Delta) - F(\theta - c\Delta)}{2c\Delta_j} \tag{14}$$

where $c$ is a step size constant and $\Delta \in \mathbb{R}^M$ is a size $M$ random variable with independent, zero-mean, bounded second moments, and bounded inverse moments, i.e. $\mathbb{E}(|\Delta|_j^{-1})$ is uniformly bounded for all

$j$. A common choice for $\Delta$ is a Bernoulli random variable with equal probabilities of being $+1$ or $-1$ for every entry.

Consider a special case, $F$, for pedagogical purposes such that the gradient at the point $\theta$ is a constant $g$ along all coordinates, the function is nearly linear at the point examined, and the number of coordinates $M$ is large in a central limit theorem sense. We then have, $F'(\theta)_j = g$ for all $j$, and $F(\theta + c\Delta) \approx F(\theta) + c\ F'(\theta)^T \Delta = F(\theta) + cg\ \vec{1}^T \Delta \approx F(\theta) + \mathcal{N}(0, cgM)$. On a quantum computer, the estimator will be constructed by taking independent measurements of $F(\theta \pm c\Delta)$ and then rescaling the sample mean by $1/2c\Delta_j$. We then see that the variance of an individual term in this case is given by

$$\text{Var}[\bar{F}'(\theta)_j] = \frac{F(\theta)}{c} + gM \tag{15}$$

As such the number of samples required to reach a precision $\epsilon$ with high probability in even a single gradient component scales as

$$N_s = \frac{F(\theta)/c + gM}{\epsilon^2} \tag{16}$$

which clearly increases linearly with the number of components $M$, and does not achieve the desired scaling despite the estimator being constructed from only two function calls. It is also worth noting that the estimates for each component of the gradient are highly correlated across the vector, which can lead to larger errors than would be otherwise expected under alternative norms. This is intuitively expected, as it should not generally be possible to determine $M$ independent random variables from a single value without increasing the precision of the estimates at least proportionately. We note in passing that generally to obtain an unbiased estimator one must also take $c$ to be on the order of $\epsilon$, but this dependence can be improved with higher order formulas to $\epsilon^{-k}$ for some $k > 1$ [Spall, 2000], but this is not central to our study.

## B    Classical backpropagation in quantum circuits

In order to frame the discussion, it is worth considering a number of closely related setups as they would appear if performed on a quantum computer. In particular, in similar notation and cost models, its interesting to consider how classical backpropagation would look in a quantum circuit for a deterministic classical function and perhaps the closer classical analog, classical parameterized Markov processes on the space of probabilistic bits.

### B.1    Classical functions

First we will look at an entirely classical function using reversible arithmetic for the purposes of analogy, using a simplified function but with simple generalizations available. This will be helpful for setting the stage in terms of notation and scaling, and also help make a connection with the gradient algorithm of Jordan [2005]. Consider a classical function $f$ that depends on some set of parameters $x \in \mathbb{R}^M$ via more elementary functions $f_i$. For this example, we assume a simple dependency graph for the overall function $f : \mathbb{R}^M \to \mathbb{R}$ is the simple composition of elementary functions, $f = f_N \circ f_{N-1} \circ ... \circ f_1$. Given this structure, we denote a set of intermediate variables $z_i$, such that $z_i = x_i$ for $i \in [1, M]$ and $z_i = f_i(z_{\alpha(i)})$ for $i \in [M + 1, n]$ where $\alpha(i)$ is the subset of variables needed to evaluate $f_i$, noting that we are implicitly including a trivial set of elementary functions $f_i$ that are simply the identity operation. We also assume that no $z_i$ depends on itself, each $z_i$ appears exactly once, and derivatives of the elementary operations are readily available, that is a simple function for evaluating $f_i'(z)$ is available for any input $z$.

Given these definitions, we are ready to describe the algorithm for obtaining the gradient $\nabla_x f(x)$. We consider a universal precision $\delta$ for all parameters and function values, such that classical numbers use $O(\log(1/\delta))$ qubits for their representation. For initialization, we store each of the parameters $x_i$ in their own quantum register $|\rangle_x$ to run the circuit fully within the quantum computer. In the first step, we run the function evaluation in the so-called forward pass and store the intermediate values $z_i$ each in their own quantum register $|\rangle_z$ using the elementary implementations of $f_i$ as reversible circuits. Taking now an additional set of auxiliary registers, $|\rangle_\lambda$ with the same size as the intermediate variables, we assign $\lambda_n = 1$, and compute the backwards pass according to reversible

implementations of $\lambda_j = \sum_{i \in \beta(j)} \partial z_j f_i(z_{\alpha(i)})$ where $\beta(i)$ is the outgoing nodes for intermediate variables $z_i$. In the final step, we may simply read off the $\lambda$ register to find $\nabla_x f(x) = \lambda_{1:M}$.

Considering a general auxiliary register $|\rangle_A$, these steps may be written in quantum form as

$$|x\rangle_x |0\rangle_z |0\rangle_\lambda |0\rangle_A \xrightarrow{\text{Forward}} |x\rangle_x |z\rangle_z |0\rangle_\lambda |r_f\rangle_A \xrightarrow{\text{Backward}} |x\rangle_x |z\rangle_z |\lambda\rangle_\lambda |r_b\rangle_A \qquad (17)$$

where $r_f$ and $r_b$ denote the state of the arithmetic trash register after the forward and backwards pass respectively. Given our precision specification, the size of each of the $x$ register is $\tilde{O}(M)$ and the size of the $z$ and $\lambda$ registers are $\tilde{O}(N)$. This representation is a bit wasteful in that as the backwards pass proceeds one can overwrite the intermediate values $z$ with $\lambda$ when they are no longer needed, but writing it this way clarifies the steps. If we assume a typical setup where the number of free parameters is roughly on par with the number of elementary functions, then we see that the total storage for the primary registers is $\tilde{O}(M)$ and similar for the ancillary register. Similarly, the amount of computation required in both the forward and backwards pass is $\tilde{O}(M)$, or approximately twice the cost of evaluating the function in the forward direction, meeting the scaling requirements of backpropagation with some small overhead for maintaining reversibility.

It is useful to compare some aspects of this approach to the quantum algorithm of Jordan for evaluating gradients of classical functions using a single black box function query [Jordan, 2005]. Considering only the computation, if we approximate the forward pass and backwards pass to each be the same cost as one black box function query, then up to log factors in precision of evaluation this method is a constant factor of two more expensive. Said another way, there is no quantum advantage in evaluating the gradient when one has white box access to the classical function implementation and it satisfies the simple dependencies requirements. In terms of storage requirements, the algorithm of Jordan requires the same $x$ register, but makes no use of the intermediate variable registers such as $z$ or $\lambda$ (which can be combined in real implementations to be approximately the size of the $x$ register). This use of intermediate storage is sometimes characterized as a form of dynamic programming, where the storage of intermediate variables reduces overall computational complexity. Moreover, this version takes advantage of analytical gradients of the subfunctions which can be evaluated to high precision more easily than depending on the finite difference formulations of gradient algorithms as in Jordan's technique.

So in summary, both a quantum implementation of classical backpropagation and Jordan's technique have a computational cost that is constant in the number of parameters if our cost model considers overall function evaluations as the cost model. This represents an exponential improvement over naive finite difference computations or symbolic evaluation of derivatives one element at a time. The backpropagation technique utilizes an extra storage register and knowledge of the problem structure, as is common in dynamic programming, while Jordan's algorithm needs only black-box queries. Both of the techniques assume bitwise access to the oracle as a classical function.

### B.2 Classical parameterized Markov chains

In the previous section, the comparison of classical backpropagation and Jordan's algorithm made use of bitwise access to a classical, deterministic function. The case of a classical function encoded in bits helps frame the discussion in not only scaling but also the sense in which classical parameterized functions are perhaps not the best analog for parameterized quantum circuits. A key aspect of this difference was highlighted in Gilyén et al. [2019] by showing that in the black box setting, it was more appropriate to consider current parameterized quantum circuits as a phase or amplitude oracle, in which case they prove a lower bound of at least $M^{1/2}$ calls to the black box (in contrast to $O(1)$), ruling out the desired backpropagation scaling except for special cases. This contrast motivates asking whether the intuitive origin of this lower bound is related more to the black box nature of the access, the quantum nature of the parameterization, or merely the probabilistic features of the parameterization. Here we show that a classical analog to parameterized quantum circuits, namely parameterized Markov processes do indeed allow the analog of classical backpropagation which helps highlight that the difficulty in achieving constant scaling is due to the quantum nature of the problem.

To draw an analogy between quantum and probabilistic classical states for our purposes, we will introduce a small number of analogous concepts that are considered in greater depth by Baez and Biamonte [2012]. A parameterized quantum state $|\psi(\theta)\rangle$ is an $L^2$ normalized state such that

$\int_S ds \ |\psi(s;\theta)|^2 = 1$, that is often formulated as a parameterized quantum circuit acting on a known initial state as $|\psi(\theta)\rangle = U(\theta)|0\rangle$ where $U$ is a unitary transformation. In contrast, a parameterized classical probability vector $|\psi(\theta))$ is a positive $L^1$ normalized probability vector such that $\int_S ds \ \psi(s;\theta) = 1$, that may be formulated as a parameterized classical circuit acting on a known reference state as $|\psi(\theta)) = U(\theta)|0)$ where $U$ is a left-stochastic operation in this case. As a connection between the two, one may consider classical transformations as the set of transformations restricted to the diagonal of a quantum density matrix, and note that it is always possible to represent a classical probability process as a quantum process, albeit non-uniquely, but the converse is of course not true in general.

The corresponding analog of expected values of Hermitian operators on quantum states will be expected values with diagonal operators $O$. Such operators are well defined for expected values on both classical and quantum states and are identical when the quantum populations are equal to the classical probabilities. In setting up for the computation of gradients with respect to the parameter vectors, we will consider objective functions defined by the same observable $O$ and a sequence of operations that each depend on a single parameter. That is, the corresponding classical and quantum objectives with these assumptions may be concisely defined by

$$f(\theta) = \int_S ds \ O(s)|(\prod_i U_i(\theta_i)\psi^0)(s)|^2 = \langle O \rangle_{U(\theta)\psi^0} \tag{18}$$

$$f(\theta)_c = \int_S ds \ O(s)(\prod_i U_i(\theta_i)\psi_c^0)(s) = \langle O \rangle_{U(\theta)\psi_c^0}. \tag{19}$$

Our question here will be if the restriction to parameterized classical stochastic processes allows the desired scaling in determining gradients of an expected value with the given parameters. The evaluation of gradients with respect to parameters in quantum circuits relies largely on the fact that anti-Hermitian operators generate unitary evolutions, and we may exploit that relationship to determine gradients as expected values explicitly. There is a direct analogy to this for general stochastic operators, in that they are generated by so-called infinitesimal stochastic operators, defined by $\sum_i H_{ij} = 0$. With this definition, in finite dimensions they characterize the family of Markov semi-groups via exponentiation as $U(t) = \exp(Ht)$. For our purposes, it suffices that this yields a well defined operator for evaluation of single parameter derivatives.

In order to properly compare the two settings, we need to make clear a number of assumptions on the operators $U_i$ and corresponding operators $H_i$ that mirror assumptions in the quantum case, allowing efficient implementation. To begin, we assume each $U_i(\theta_i)$ is a simple operation, analogous to a quantum gate or Pauli operator, such that it is defined as a tensor product on a classical probabilistic bit space, and evaluating the transition probability between two basis states is efficient to do at high precision. In general, the basis could change between steps and the process could remain efficient, however for simplicity we consider the standard computational basis here. Moreover, we assume that the operation that generates the $U_i$, which we denote $H_i$ is simple to evaluate between basis states, and has a bounded norm $||H_i|| = 1$, so that parameters $\theta_i$ have consistent and reasonable scales. Similarly, we will restrict ourselves to observables $O$ with reasonable norms, i.e. $||O|| = 1$.

With these assumptions, we investigate derivatives of a classical stochastic process under different sampling schemes. Let's imagine we have a stochastic process $U$, much like a variational circuit, which we write as

$$U(\theta) = \prod_i U_i(\theta_i) \tag{20}$$

where each $U_i$ is a stochastic process with a corresponding generator $H_i$, such that

$$U_i(\theta_i) = \exp(\theta_i H_i) \tag{21}$$
$$\partial_{\theta_i} U_i(\theta_i) = H_i U_i(\theta_i) \tag{22}$$

We will be sampling the expected value of some observable $O$ which is a diagonal matrix in our construction, and so the function value we are interested in optimizing, given a initial probability distribution $\psi_0$ can be written in a number of ways, but some are

$$f(\theta) = \langle O \rangle_{U(\theta)\psi_0} \tag{23}$$

$$= \int OU(\theta)\psi_0 \tag{24}$$

Now if we take the gradient of this function with respect to the parameters, we find

$$\partial_{\theta_i} f(\theta) = \partial_{\theta_i} \langle O \rangle_{U(\theta)\psi_0} \tag{25}$$

$$= \int O \prod_{j<i} U_j \partial_{\theta_i} U_i \prod_{k>i} U_k \psi_0 \tag{26}$$

$$= \int O \prod_{j<i} U_j H_i U_i \prod_{k>i} U_k \psi_0. \tag{27}$$

Using this construction, one can store the trajectory and lean on a path-integral formalism to use a single sampling process to take independent samples of all the gradient components with each stochastic sample that is taken. One way to write this is to borrow the path-integral like formalism using resolutions of the identity as

$$f(\theta) = \int O \prod_j U_j \psi_0 \tag{28}$$

$$= \sum_{i_1,...,i_N} \int O \ket{i_N} \bra{i_N} U_N \ket{i_{N-1}} \bra{i_{N-1}} U_{N-1} ... \bra{i_1} \psi_0$$

$$= \sum_{i_1,...,i_N} p(i_1, ..., i_N) O(i_N)$$

where we use $p(i_1, ..., i_N)$ to represent the probability of a particular configuration that was sampled, and similarly $O(i_N)$ for the value of the final configuration. We assume that for each individual configuration it is possible to compute the transition probability between individual configurations, e.g. $\bra{i_N} U_N \ket{i_{N-1}}$ which is typically true in the classical case as well. As a result, for a given path, we use re-weighting to make that path produce an unbiased sample for the gradient component we are interested in as well. In particular, writing the same for the shifted gradient estimator for component $j$ merely requires substituting the relevant matrix element

$$\bra{i_j} U_j \ket{i_{j-1}} \to \bra{i_j} H_j U_j \ket{i_{j-1}} \tag{29}$$

hence we can estimate the gradient using samples re-weighted by

$$\partial_{\theta_j} f(\theta) = \sum_{i_1,...,i_N} p(i_1, ..., i_N) \left( \frac{\bra{i_j} H_j U_j \ket{i_{j-1}}}{\bra{i_j} U_j \ket{i_{j-1}}} \right) O(i_N) \tag{30}$$

where the weighting factors we also assume to be efficiently computable by construction of the elementary operations $U_i$, which is analogous to the quantum generators typically used as well, defined as simple operations lifted into large spaces by tensor products. This suggests the following procedure for efficiently estimating gradients with respect to parameters in the classical analog of quantum variational circuits.

1. Draw a sample from $\psi^0$ and store this configuration as $\ket{i_i}$, which may be represnted efficiently as a classical bit string.

2. For each elementary operation $U_i$, sample the next classical configuration with probability determined by $U_i$, and store the configuration as $\ket{i_j}$.

3. Upon reaching the final configuration, evaluate $O(i_N)$ from the definition of $O$ to determine the value of the objective.

4. Using the stored path, $\{\ket{i_j}\}$, for each elementary step, sample $\left( \frac{\bra{i_j} H_j U_j \ket{i_{j-1}}}{\bra{i_j} U_j \ket{i_{j-1}}} \right) O(i_N)$ and store the value in a vector to be used in a running average that determines the gradient.

5. Repeat this procedure until the uncertainty in the estimate for each gradient component is as low as desired.

It is easy to see from the above procedure that the variance in the estimate of each individual gradient component does not have an explicit dependence on the number of elementary steps. This can be seen from Equation (30), which only has an explicit dependence on 3 points in the chain. Alternatively, from our assumptions designed to mirror the case of quantum circuits, we know the variance of

these estimators is controlled by the value of the product $(i_j|\, H_j U_j\, |i_{j-1})\, O(i_N) \leq 1$, independent of the number of parameters or steps in the sampling process. It may appear that the quantity estimated could be unbounded, but if we move the denominator into $p$, the result is again a probability distribution multiplied only by values determined by the numerator here. As a result, analogous to backpropagation in the bitwise function case, by storing the intermediate configurations $\{|i_j\rangle\}$ at a cost memory of $O(M)$, we see that evaluating the gradient requires a number of samples that is independent of $M$.

From this, we see that indeed the desired scaling is possible in the case of the analogous classical parameterized stochastic processes on tensor product spaces. The formulation as a sum over paths also allows us to make connection to the gentle measurement results in the main text, in that we are always promised to be in a computational basis state, making it possible to do a gentle measurement at intermediate steps with unit probability. This division allows us to help identify the origin of challenges in achieving backpropagation scaling as a problem with quantum measurement collapse and the inability to read out intermediate states while continuing a computation, rather than the probabilistic formulation of the problem. In addition, one may make the classical generators $H_i$ non-commutative with each other and suffer no additional difficulties in estimating the gradient components, unlike in the quantum case. It remains an interesting question to better understand the performance separation on practical tasks between quantum variational methods and this type of classical analog, given the advantage in trainability of the classical construction.

## C   Polynomial complexity circuits

It is reasonable to ask if we can first rule out backpropagation when only given access to single copies of a state. A useful tool to rule out the possibility of certain tasks is information-theoretic bounds, however, we show here that these are not sufficient to rule out quantum backpropagation scaling on single copies as the task remains information-theoretically viable under the assumption of a polynomial length variational circuit, thanks to classical shadows. On the other hand, standard computational arguments illustrate the difficulty in acheiving the desired scaling.

### C.1   Information-efficiency with classical shadows

The idea behind classical shadows is to create a classical representation of a state $\rho$, that allows one to affordably estimate other properties of interest, like expectation values of observables [Huang et al., 2020]. In general, the number of samples, $N$, needed to predict say, $\mathrm{Tr}[E_1 \rho], ..., \mathrm{Tr}[E_K \rho]$ within additive error $\varepsilon$, with high probability is

$$N = \Omega(\log(K)\, \max_i \|E_i\|_{\mathrm{shadow}}/\epsilon^2),$$

where $\|E_i\|_{\mathrm{shadow}}$ is a norm influenced by the particular measurement primitive chosen to implement the classical shadow scheme. While general quantum states can be hard to determine, the additional constraint of a state being generated by a polynomial complexity variational circuit allows us to strengthen our statements.

**Definition 15** (Polynomial complexity circuit). We say a circuit is a *polynomial complexity circuit* if it is composed from a fixed gate set $G$ that may be applied between any two qubits with a maximum number of gates scaling polynomially in $n$, the number of qubits. Additionally, we will call it a *polynomial complexity parameterized circuit* if each gate in the elementary set is defined by a bounded number of parameters.

With this at hand, we have the following.

**Proposition 16** (Information-efficiency of polynomial complexity circuits). *Let $\rho = |\psi\rangle\langle\psi|$ be the density matrix of a pure state generated from a quantum circuit of polynomial complexity built from a gate set of size $G$ applied between any two qubits, with at most $p(n)$ total gates, where $p(n)$ is a polynomial in the number of qubits, $n$. With these definitions, there are at most $K = (nG)^{2p(n)}$ of these circuits. Then, $\rho$ can be explicitly determined using $\Omega(\log(K)/\varepsilon^2) = \Omega(2p(n)\log(nG)/\varepsilon^2)$ single-copy measurements and a classical search procedure.*

*Proof.* Given that $|\psi\rangle$ is generated from a polynomial complexity circuit, denote the possible states created by such a circuit as $|\phi_i\rangle$. With the above definitions it is easy to see that the total number of

possible states that can be generated by a single step is $n^2 G$, and hence with $p(n)$ possible choices, the total number of states is $K = (nG)^{2p(n)}$. If the underlying set of operations used to generate the state is unknown, it is still possible to cover the space of two-qubit operations to diamond distance error $\epsilon$ with a number of operations scaling polynomially in $1/\epsilon$ and $p(n)$ [Caro et al., 2022]. If we denote this number of extended operations as $G'$, then the argument proceeds as before in terms of asymptotic scaling by replacing $G$ with $G'$. Performing Clifford classical shadows with $E_i = |\phi_i\rangle\langle\phi_i|$ for $i = 1, ..., K$, one can estimate the fidelity, i.e. $\text{Tr}[E_i |\psi\rangle\langle\psi|]$, for all $i$ within additive error $\varepsilon$ using $\Omega(\log(K)/\varepsilon^2)$ single copies of $|\psi\rangle$. Since $|\psi\rangle$ is generated by one of the $K$ circuits, searching for an $E_i$ that provides the maximum fidelity, allows one to find $\text{Tr}[E_i |\psi\rangle\langle\psi|] = 1$, with high probability, and thus, explicitly determine $|\psi\rangle$, and a circuit that generated it by using classical simulation of the family of circuits, that will generally scale both exponentially in $n$ and $K$. $\qquad\square$

With this knowledge, one may proceed to compute expectation values classically to determine gradients or indeed any desired expected value or feature of the state. Whilst this procedure allows us to determine $|\psi\rangle$ and a circuit for creating it, executing it incurs quantum hardware costs dominated by the Clifford circuits needed for the classical shadow protocol – which are of polynomial depth, but contain entangling gates which are limiting in practice. Even more concerning, is the classical cost of post-processing. Obtaining the maximum fidelity involves storing $K = (n + p(n))^{O(p(n))}$ values and searching over them, which can be expensive. Additionally, the final computation of the expectation values needed for backpropagation, requires knowing and storing $M$ exponentially large matrices, over and above the cost to compute the expectation values. And so, backpropagation scaling remains untenable with this implementation.

### C.2  Computational hardness on polynomial complexity circuits

The result and algorithm (a brute force search) used in Proposition 16 demonstrate the information-theoretic efficiency of determining almost anything one would want to know about a state if we are guaranteed that it is both a pure state and generated by a polynomial complexity circuit. The classical computational procedure is clearly inefficient, but this begs the question of whether an efficient procedure might exist in general, especially given the existence of an efficient procedure for special cases. Here we argue that no efficient procedure can exist in the most general case, unless it is possible to efficiently clone pseudo-random quantum states.

**Proposition 17** (Computational hardness of polynomial complexity circuits). *Under standard cryptographic assumptions, no efficient computational procedure exists to identify a pure state of polynomial complexity to trace distance $\varepsilon$.*

*Proof.* A pseudo-random quantum state is defined to be a pure state of polynomial complexity that no efficient computational algorithm given a polynomial number of copies of the state can distinguish from the Haar random state. Using the procedure described in Proposition 16, a circuit that can recreate the state to trace distance $\epsilon$ can be found using a polynomial number copies of the state. If the procedure that finds this circuit is also computationally efficient, then the state can be cloned efficiently, violating the no-cloning theorem for pseudo-random states shown in Ji et al. [2018], which merely rests upon standard cryptographic assumptions. $\qquad\square$

This result demonstrates that even if we know a state is a pure state generated from a polynomial complexity circuit, it is computationally infeasible to identify it under cryptographic assumptions despite the information-theoretic efficiency. This suggests that there are states and observables for which the backpropagation problem could remain challenging, and that the most effective strategies must make use of known structure in the observables and states to achieve computational efficiency in analogy to known special cases.

## D  Shadow tomography protocol for gradients

For much of this manuscript it has been assumed that one has complete white-box access to the input state $\rho = |\psi(\theta)\rangle\langle\psi(\theta)|$. In a more traditional quantum setting, however, this may not be the case. One may be given access to unknown quantum states, or partially unknown states, and tasked to process them for some machine learning task. In such an instance, the input states are usually referred to as

quantum data, and insights pertaining to this model set up can be found in Huang et al. [2021]. In this section, we discuss some details around this model type, which we call a quantum neural network and is defined in Definition (8).

## D.1 Gradients as observables

Before presenting our algorithm for performing quantum backpropagation, we begin with the following remark on quantum neural networks which allows us to exploit a shadow tomography procedure.

**Remark 18** (Gradient of a quantum neural network). The $k^{\text{th}}$ gradient component of the quantum neural network may be expressed as

$$\partial_{\theta_k} \text{QNN}_{\vec{\theta}}(|\varphi\rangle) = 2 \operatorname{Re} \langle 0|\langle \varphi|\mathcal{U}^\dagger(\vec{\theta}) Z_0 \partial_{\theta_k} \mathcal{U}(\vec{\theta})|0\rangle|\varphi\rangle$$
$$= 2 \operatorname{Re} \langle \Phi_k|\Psi_k\rangle$$

where

$$|\Psi_k\rangle = (iP_k)e^{i\theta_k P_k} U_k \ldots e^{i\theta_1 P_1} U_1 |0\rangle|\varphi\rangle$$
$$= e^{i(\theta_k + \frac{\pi}{2})P_k} U_k \ldots e^{i\theta_1 P_1} U_1 |0\rangle|\varphi\rangle$$
$$|\Phi_k\rangle = U_{k+1}^\dagger e^{-i\theta_{k+1} P_{k+1}} \ldots U_M^\dagger e^{-i\theta_M P_M} Z_0 e^{i\theta_M P_M} U_M \ldots e^{i\theta_1 P_1} U_1 |0\rangle|\varphi\rangle.$$

If one defines

$$\mathcal{U}_k^{(\Psi)} = e^{i(\theta_k + \frac{\pi}{2})P_k} U_k \ldots e^{i\theta_1 P_1} U_1,$$
$$\mathcal{U}_k^{(\Phi)} = U_{k+1}^\dagger e^{-i\theta_{k+1} P_{k+1}} \ldots U_M^\dagger e^{-i\theta_M P_M} Z_0 e^{i\theta_M P_M} U_M \ldots e^{i\theta_1 P_1} U_1,$$

then, given a copy of $|\varphi\rangle$, one may attach an ancilla qubit labelled $*$ in the $|+\rangle$ state (in addition to the output qubit 0). In doing so, consider applying control-$\mathcal{U}_k^{(\Psi)}$ conditional on the ancilla being $|0\rangle$, and control-$\mathcal{U}_k^{(\Phi)}$ conditional on the ancilla being $|1\rangle$. This produces the state

$$\frac{1}{\sqrt{2}} \big( |0\rangle|\Psi_k\rangle + |1\rangle|\Phi_k\rangle \big).$$

Measuring $X$ on the ancilla qubit, the expectation is

$$\frac{1}{2} \big( \langle 0|\langle \Psi_k| + \langle 1|\langle \Phi_k| \big) X_* \big( |0\rangle|\Psi_k\rangle + |1\rangle|\Phi_k\rangle \big) = \operatorname{Re} \langle \Phi_k|\Psi_k\rangle$$
$$= \frac{1}{2} \partial_{\theta_k} \text{QNN}_{\vec{\theta}}(|\varphi\rangle).$$

This implicitly gives an operator on $|+\rangle|0\rangle|\varphi\rangle$ whose expectation value is $\frac{1}{2}\partial_{\theta_k}\text{QNN}_{\vec{\theta}}(|\varphi\rangle)$. Moreover, we can implement this measurement with $O(M)$ quantum operations.

## D.2 Proof of Theorem 9

In order to prove Theorem 9, we need to discuss and modify two concepts: online learning and threshold search [Aaronson et al., 2018, Bǎdescu and O'Donnell, 2021].

### D.2.1 Online learning of quantum states

As in Aaronson et al. [2018], suppose we have access to a stream $(E_1, b_1), \ldots, (E_M, b_M)$ where each $b_k = \langle \psi|E_k|\psi\rangle$. We want to compute hypothesis states $\omega_1, \ldots, \omega_M$, which are mixed states stored in classical memory, such that

- $\omega_k$ depends only on $(E_1, b_1), \ldots, (E_{k-1}, b_{k-1})$ (the online condition)
- $|\operatorname{Tr}(E_k \omega_k) - \langle \psi|E_k|\psi\rangle| > \varepsilon$ for as few $k$ as possible

One may produce the following theorem.

**Theorem 19.** *[Aaronson et al., 2018, Theorem 1] In the above setting, there is an explicit strategy for outputting hypothesis states $\omega_1, \ldots, \omega_M$ such that $|\operatorname{Tr}(E_k \omega_k) - \langle \psi|E_k|\psi\rangle| > \varepsilon$ for at most $O(\frac{n}{\varepsilon^2})$ values of $k$. This holds even if the measurements $b_k$ are noisy, and only satisfy $|b_k - \langle \psi|E_k|\psi\rangle| \leq \frac{\varepsilon}{3}$*

Two remarks are in order: first, the problem setup and algorithm presented in Theorem 19 are both completely classical. Second, this theorem says nothing about computational runtime. Implementation of the algorithm in Theorem 19 using techniques from convex optimization will require runtime polynomial in the dimension of the Hilbert space $\mathrm{poly}(2^n)$.

### D.2.2 Quantum Threshold Search

Bǎdescu and O'Donnell [2021] promote online learning to a shadow tomography protocol using a procedure which they call *threshold search*. This gives an improved version of the quantum private multiplicative weights algorithm proposed in Aaronson and Rothblum [2019]. The difference between the online learning setting from the previous section and general shadow tomography, is that in practice, we are typically *not* given the expectation values $\{b_k\}$ and must measure them ourselves. This is where threshold search comes in handy. Suppose we possess some copies $|\psi\rangle^{\otimes m}$ of a quantum state and are given a stream $(E_1, a_1), \ldots, (E_M, a_M)$ where each $a_k$ is supposed to be a guess such that $a_k \approx \langle \psi | E_k | \psi \rangle$. Threshold search is a subroutine which, given only logarithmically many copies of the state, can check in an online fashion whether there is an $a_k$ which errs by more than $\varepsilon$. More formally, we have the following theorem.

**Theorem 20.** *[Bǎdescu and O'Donnell, 2021, Lemma 5.2] Given $m$ copies of an $n$-qubit quantum state $|\psi\rangle^{\otimes m}$, $M$ observables $-1 \leq E_1, \ldots, E_M \leq 1$, and guesses $a_1, \ldots, a_M$, there is an algorithm which outputs either*

- $|a_k - \langle \psi | E_k | \psi \rangle| \leq \varepsilon \; \forall k$.

- *Or $|a_k - \langle \psi | E_k | \psi \rangle| > \frac{3}{4}\varepsilon$ when in fact $|b_k - \langle \psi | E_k | \psi \rangle| \leq \frac{1}{4}\varepsilon$ for a particular $k$ and value $b_k$.*

*It does so using number of copies only*

$$m = O\left( \frac{\log^2 M}{\varepsilon^2} \right).$$

*Furthermore, the algorithm is online in the sense that:*

- *The algorithm is initially given only $M$ and $\varepsilon$. It then selects $m$ and obtains $|\psi\rangle^{\otimes m}$.*

- *Next, observable/threshold pairs $(E_1, a_1), (E_2, a_2), \ldots$ are presented to the algorithm in sequence. When each $(E_k, a_k)$ is presented, the algorithm must either 'pass', or else halt and output $|a_k - \langle \psi | E_k | \psi \rangle| > \frac{3}{4}\varepsilon$.*

- *If the algorithm passes on all $(E_k, a_k)$ pairs, then it ends by outputting $|a_k - \langle \psi | E_k | \psi \rangle| \leq \varepsilon \; \forall k$*

We stress that this subroutine requires quantum memory and multi-copy measurements, and uses gentle measurements in an essential way. One is able to check whether or not $a_k$ is inside the threshold without greatly disturbing the copies of the quantum state. We are now ready to state the full shadow tomography protocol from Bǎdescu and O'Donnell [2021]. The idea is to run the online learning algorithm from Theorem 19 in parallel with threshold search, and Bǎdescu and O'Donnell [2021, Theorem 1.4] tells us that this algorithm succeeds in outputting estimates $|b_k - \langle \psi | E_k | \psi \rangle| \leq \varepsilon$ with high probability.

When applying Algorithm 1 to the observables corresponding to gradients described in Appendix D.1, we can exploit that the observables are related sequentially. In between each round $k$, we rotate both, the states stored in quantum memory and the classical online learner, so that implementing the measurement of the next gradient only requires runtime independent of $M$. Since these rotations are unitary and do not reduce the quality of any approximations, the same proof as Bǎdescu and O'Donnell [2021, Theorem 1.4] will apply. This establishes Theorem 9.

By Bǎdescu and O'Donnell [2021, Theorem 1.4], this algorithm obtains estimates $|b_k - \frac{1}{2}\partial_{\theta_k}\mathrm{QNN}_{\vec{\theta}}(|\varphi\rangle)| \leq \varepsilon$ for each $k$ by taking the number of copies to be

$$m = O\left( \frac{n \log^2 M}{\varepsilon^4} \right).$$

---
**Algorithm 1** Online and gentle shadow tomography
---

**Input:** $m$ copies of the unknown input state $|\psi\rangle^{\otimes m}$, in $m$ registers each with $n$ qubits.
**Output:** Estimates $b_k \approx \langle\psi|E_k|\psi\rangle$

1. Set $R = O(\frac{n}{\varepsilon^2})$ and $m_0 = O(\frac{\log^2 M}{\varepsilon^2})$. We need $R$ batches, each with $m_0$ copies, so $m = Rm_0$ copies in total. This gives in total

$$m = O\left(\frac{n\log^2 M}{\varepsilon^4}\right)$$

2. Initialize the online learner $\omega_1$ according to the online learning algorithm.

3. Start with the first batch of copies $|\psi\rangle^{\otimes m_0}$.

4. For each $k = 1, \ldots, M$:

   (a) Use the online learner to predict $a_k = \text{Tr}(E_k\omega_k)$.

   (b) Use threshold search to check $|a_k - \langle\psi|E_k|\psi\rangle|$.

   (c) If threshold search passes $|a_k - \langle\psi|E_k|\psi\rangle| \le \varepsilon$,

      i. Output estimate $b_k \leftarrow a_k$.

      ii. Leave the online learner unchanged $\omega_{k+1} \leftarrow \omega_k$.

   (d) If threshold search concludes $|a_k - \langle\psi|E_k|\psi\rangle| > \frac{3}{4}\varepsilon$ and in fact $|b_k - \langle\psi|E_k|\psi\rangle| \le \frac{1}{4}\varepsilon$,

      i. Output estimate $b_k$.

      ii. Update online learner with $b_k \approx \langle\psi|E_k|\psi\rangle$ to get $\omega_{k+1}$.

      iii. Discard the current batch and move onto a fresh batch $|\psi\rangle^{\otimes m_0}$.

---

Moreover, the required number of quantum operations is

$$O(mM) = O\left(\frac{nM\log^2 M}{\varepsilon^4}\right)$$

This is quasi-linear in $M$. With naive storage of the entire density matrix of the hypothesis state $\omega_k$, the classical cost is

$$M \cdot 2^{O(n)}$$

Which is also linear in $M$, but unfortunately exponential in the input size $n$. We present the full algorithm for gradient estimation using online shadow tomography with threshold search in Algorithm 2.

**Algorithm 2** Shadow tomography protocol for gradients of a quantum neural network

---

**Input:** $m$ copies of the unknown input state $|\varphi\rangle^{\otimes m}$ in $m$ registers each with $n$ qubits.
**Output:** Estimates $b_k \approx \frac{1}{2}\partial_{\theta_k}\mathrm{QNN}_{\vec{\theta}}(|\varphi\rangle)$ for $k = 1, \ldots, M$

1. Set $R = O(\frac{n}{\varepsilon^2})$ and $m_0 = O(\frac{\log^2 M}{\varepsilon^2})$. We need $R$ batches, each with $m_0$ copies, so $m = Rm_0$ copies in total. This gives

$$m = O\left(\frac{n\log^2 M}{\varepsilon^4}\right)$$

2. Define for each $k = 1, \ldots, M$

$$|\psi_k\rangle = \frac{1}{\sqrt{2}}\big(|0\rangle|\Psi_k\rangle + |1\rangle|\Phi_k\rangle\big)$$

and recall from Remark 18 that

$$\langle\psi_k|X_*|\psi_k\rangle = \frac{1}{2}\partial_{\theta_k}\mathrm{QNN}_{\vec{\theta}}(|\varphi\rangle)$$

3. Attach the output qubit and an ancilla qubit in the $|+\rangle$ state to each register. Label the output qubit $0$ and the ancilla qubit $*$.

4. To each register, do the following:

   (a) Apply control-$\mathcal{U}_1^{(\Psi)}$ conditional on the ancilla being $|0\rangle$. This requires $O(1)$ quantum operations.

   (b) Apply control-$\mathcal{U}_1^{(\Phi)}$ conditional on the ancilla being $|1\rangle$. This requires $O(M)$ quantum operations. This step is analogous to the initial forward pass in classical backpropagation.

   (c) This produces the state $|\psi_1\rangle^{\otimes m}$.

5. Initialize the online learner $\omega_1$ according to the online learning algorithm.

6. Start with the first batch of copies $|\psi_1\rangle^{\otimes m_0}$

7. For $k = 1, \ldots, M$, do the following. This loop is analogous to the backward pass in classical backpropagation.

   (a) Use the online learner to predict $a_k = \mathrm{Tr}(X_*\omega_k)$.

   (b) Use threshold search to check $|a_k - \langle\psi_k|X_*|\psi_k\rangle|$. This takes time independent of $M$.

   (c) If threshold search passes $|a_k - \langle\psi_k|X_*|\psi_k\rangle| \leq \varepsilon$,

      i. Output estimate $b_k \leftarrow a_k$.

      ii. Leave the online learner unchanged $\omega_{k+1} \leftarrow \omega_k$.

   (d) If threshold search concludes $|a_k - \langle\psi_k|X_*|\psi_k\rangle| > \frac{3}{4}\varepsilon$ and in fact $|b_k - \langle\psi_k|X_*|\psi_k\rangle| \leq \frac{1}{4}\varepsilon$,

      i. Output estimate $b_k$.

      ii. Update online learner with $b_k \approx \langle\psi_k|X_*|\psi_k\rangle$ to get $\omega_{k+1}$.

      iii. Discard the current batch and move onto a fresh batch.

   (e) To each register in the current batch *and the unused batches*, do the following:

      i. Apply control-$(e^{i(\theta_{k+1}+\frac{\pi}{2})P_{k+1}}U_{k+1}e^{-i\frac{\pi}{2}P_k})$ conditional on the ancilla being $|0\rangle$. This implements $\mathcal{U}_{k+1}^{(\Psi)}(\mathcal{U}_k^{(\Psi)})^{-1}$, and only requires $O(1)$ quantum operations.

      ii. Apply control-$e^{i\theta_{k+1}P_{k+1}}U_{k+1}$ conditional on the ancilla being $|1\rangle$. This implements $\mathcal{U}_{k+1}^{(\Phi)}(\mathcal{U}_k^{(\Phi)})^{-1}$, and only requires $O(1)$ quantum operations.

      iii. This produces in each batch (a noisy approximation to) the state $|\psi_{k+1}\rangle^{\otimes m_0}$.

   (f) Also apply the rotations in Step (e) to the hypothesis state $\omega_{k+1}$ in classical memory. The online learner now approximates $|\psi_{k+1}\rangle\langle\psi_{k+1}|$.

---

# E    Fully gentle gradient estimation

In this section, we motivate for a need to perform sequential and gentle measurements to individual gradient states, as opposed to superpositions of them. Thereafter, we discuss general strategies based on gentle measurements alone, performed on single and multiple copies.

## E.1    Considering individual gradient states

While we briefly motivated the need for a sequential reuse of information in measurements in the main text, here we further motivate such a construction as a necessary, but perhaps not sufficient condition for our purposes. Given that one can create a superposition over all the potential gradient components at a cost that only requires a single function call, it is natural to ask if this ability gives us any headway in achieving our goals. Consider exploiting the superposition over all gradient states

$$|\Psi\rangle = \sum_{k=1}^{M} c_k |A_k\rangle \prod_{j\in A} U_j |0\rangle = \sum_{k=1}^{M} c_k |A_k\rangle |\psi_k\rangle, \tag{31}$$

using at most $cM$ calls to the family $\{U_j\}$ and some ancillary qubits $|A_k\rangle$ associated with the $k^{\text{th}}$ gradient state.[2] Creating such a superposition weakens our ability to extract each gradient component's signal upon measurement, and thus, requires more samples to distinguish between gradient components with a desired precision. From a cost perspective, it remains optimal or equivalent to consider gradient states $|\psi_k\rangle$ individually. To make this more concrete, consider a state discrimination task, with the following lemma at hand.

**Lemma 21** (Optimal two-state discrimination). *Any quantum algorithm that distinguishes two states $\rho_1$ and $\rho_2$ using a single copy of each state with probability at least $0.9$ requires*

$$\frac{1}{2} + \frac{1}{2}\|\rho_1 - \rho_2\|_{\text{tr}} \geq 0.9. \tag{32}$$

Now we may proceed to the state discrimination task, where it is clear a superposition is not helpful.

**Proposition 22.** *Consider the two-state discrimination task for two scenarios. First, given $|\psi_m\rangle$ and $|\phi_m\rangle$, where $\langle\psi_m|\phi_m\rangle = 0$, there is a measurement strategy that can distinguish the states with a single measurement. Second, given the states*

$$|\Psi\rangle = \frac{1}{\sqrt{M}} \sum_{k=1}^{M} |A_k\rangle |\psi_k\rangle, \tag{33}$$

*and*

$$|\Phi\rangle = \frac{1}{\sqrt{M}} \sum_{k=1}^{M} |A_k\rangle |\phi_k\rangle, \tag{34}$$

*where $|\psi_k\rangle = |\phi_k\rangle$ for every $k$ except the $m^{th}$ component and $\langle\psi_m|\phi_m\rangle = 0$ as before, then $\Omega(M)$ copies are required by any strategy aiming to discriminate $|\Psi\rangle$ from $|\Phi\rangle$ with reasonably high success probability.*

*Proof.* The first scenario follows straightforwardly from Lemma (21) since there is no overlap between $|\psi_m\rangle$ and $|\phi_m\rangle$ – hence, their trace distance is 1 and Equation (32) always holds. For states in uniform superposition over all $M$ components, the overlap is $1 - 1/M$ which is close to unity for large $M$, indicating the difficulty of the task when the states mostly overlap. Given access to $N$ copies of $|\Psi\rangle$ and $|\Phi\rangle$, to discriminate with probability at least 0.9 requires

$$\frac{1}{2} + \frac{1}{2}\sqrt{1 - |\langle\Psi|\Phi\rangle|^{2N}} \geq 0.9, \tag{35}$$

or equivalently

$$\left(1 - \frac{1}{M}\right)^{2N} \leq 0.36, \tag{36}$$

implying that $N = \Omega(M)$ in order to discriminate successfully with the desired probability.     □

---

[2] $c$ is some small constant.

From Proposition (22), we have the immediate corollary.

**Corollary 23.** *It is either optimal or equivalent in cost to consider gradient states individually, as opposed to a superposition over them all.*

*Proof.* Replacing the uniform superposition in Equations (33) and (34) to the more general, $|\Psi\rangle = \sum_{k=1}^{M} c_k |A_k\rangle |\psi_k\rangle$ and $|\Phi\rangle = \sum_{k=1}^{M} c_k |A_k\rangle |\phi_k\rangle$, the number of samples needed to discriminate the $m^{\text{th}}$ component scales as $N \sim 1/c_m^2$. Since $c_m^2 \in [0,1]$, it is clear that $c_m^2 = 1$ is optimal. If there are $M$ components, then $c_m^2 \sim 1/M$ and hence, $N \sim M$. Assuming the superposition state $|\Psi\rangle$ incurs a cost proportional to $M$, the number of samples required to differentiate between components in the wave function will imply an overall cost that scales as $M^2$. $\square$

### E.2 A case for sequential and gentle measurement

Whilst the cost equivalence presented in Corollary 23 implies no benefit from a superposition of gradient states, it also suggests that, if one is to obtain backpropagation scaling, individual gradient states must be utilized in a more resource efficient manner. Drawing inspiration from backpropagation, if one could instead use the state $|\psi_k\rangle$ to make a measurement, then update it to $|\psi_{k+1}\rangle$ without substantially disturbing it, it would then be possible to perform all of the measurements at an overall cost scaling like $O(M)$. We illustrate such a benefit by means of an example using fictitious non-destructive measurements in Algorithm 3.

---

**Algorithm 3** Gradient estimation with a modified, non-destructive swap test

**Input:** Three registers initialized to $|+\rangle |0\rangle |0\rangle$
**Output:** Gradient vector estimate for $F(\theta)$

1. Apply $U(\theta) = U_M...U_1$ to the second register, controlled on the first being 0. Cost $\sim M$.

2. Apply $OU(\theta)$ to the third register, conditional on the first being 1. Cost $\sim M$ and the state becomes
$$|+\rangle |0\rangle |0\rangle \rightarrow \frac{1}{\sqrt{2}}(|0\rangle |\psi_M\rangle |0\rangle + |1\rangle |0\rangle |\lambda\rangle),$$
where $|\psi_M\rangle = U_M...U_1 |0\rangle$ and $|\lambda\rangle = OU_M...U_1 |0\rangle$. By assumption, all $U_j$ and $O$ are hermitian and unitary.

3. For $k$ in $\{M, M-1, ..., 1\}$:

   (a) Apply and update $|\psi_k\rangle = -iP_k |\psi_k\rangle$ conditioned on ancilla being 0. Cost $\sim 1$.

   (b) Perform a non-destructive swap test on the state
$$\frac{1}{\sqrt{2}}(|0\rangle |\psi_k\rangle |0\rangle + |1\rangle |0\rangle |\lambda\rangle)$$
   to estimate $[F'(\theta)]_{\theta_k} = -2 \operatorname{Im} \langle\lambda|\psi_k\rangle$ with no damage to the state. Cost $\sim 1$.

   (c) If $k > 1$ apply and update $|\lambda\rangle = U_k^\dagger |\lambda\rangle$ conditional on ancilla being 1. Cost $\sim 1$.

   (d) If $k > 1$ apply and update $|\psi_{k-1}\rangle = U_k^\dagger (iP_k) |\psi_k\rangle$ conditional on ancilla being 0. Cost $\sim 1$.

---

The procedure naturally breaks down in a real quantum computer at Step (3b) due to the reliance on non-destructive measurements. Substituting these for gentle measurements, which are only partially non-destructive but, at least, theoretically possible, one may still aspire to exploit the structure of the problem and achieve backpropagation scaling as in Algorithm 3.

### E.3 Gentle measurement on single copies

The need to reuse a state enough times to extract every gradient component, imposes constraints on the gentleness of measurements made. While the use of multiple copies may enhance the ability to leverage gentle measurements, it is straightforward to see why this approach would not work in general, when given access to a single copy of $\rho$. Using a scheme like the modified swap test in

Algorithm 3, implies that each measurement must be on average $1/M$-gentle in order to reuse the state $M$ times to extract each gradient component without damaging the state to the point that at least one observable on the state is completely wrong. Enforcing such a constraint, leads to measurements that are trivial – i.e. they barely depend on $\rho$ and cannot yield enough information about gradients. We recap some useful lemmas whose proofs can be found in Aaronson and Rothblum [2019] to make this more concrete.

**Lemma 24** (Additivity of damage). *Let $\rho$ be some mixed state and let $S_1, S_2, ..., S_M$ be general quantum operations. Suppose for all $j$, we have*

$$\|S_j(\rho) - \rho\|_{\mathrm{tr}} \leq \alpha_j,$$

*then*

$$\|S_M(S_{M-1}(...S_1(\rho))) - \rho\|_{\mathrm{tr}} \leq \alpha_1 + ... + \alpha_M.$$

**Lemma 25** (Trivial measurement). *Given a measurement $M$ and parameter $\eta \geq 0$, suppose that for every two orthogonal pure states $|\psi\rangle$ and $|\phi\rangle$, and every possible outcome $y$ of $M$, we have*

$$\Pr[M(|\psi\rangle) \text{ outputs } y] \leq e^\eta \Pr[M(|\phi\rangle) \text{ outputs } y].$$

*Then $M$ is $\eta$-trivial. Further, let $E_1 + ... + E_k = I$ be the POVM elements of $M$. Assume without loss of generality that the outcome $y$ corresponds to the element $E = E_1$. Then,*

$$\langle\psi| E |\psi\rangle \leq e^\eta \langle\phi| E |\phi\rangle,$$

*holds for all states, not just all orthogonal $|\psi\rangle, |\phi\rangle$.*

**Lemma 26** (Triviality lemma). *Suppose a measurement is $\alpha$-gentle on all states. Then the measurement is $\ln\left(\frac{1+4\alpha}{1-4\alpha}\right)$-trivial —so in particular, $O(\alpha)$-trivial, provided $\alpha \leq \frac{1}{4.01}$.*

Equipped with these lemmas, we proceed to demonstrate the difficulty of gentle gradient estimation with single-copy access to a pure state.

**Theorem 27.** *A sequence of $M$ measurements on a single-copy pure state that is $1/M$-gentle at every step to extract every gradient component, will be trivial.*

*Proof.* Choose a circuit such that gradient state differs substantially, i.e. $\||\psi_i\rangle\langle\psi_i| - |\psi_j\rangle\langle\psi_j|\|_{\mathrm{tr}} = 1$ for all measurements. In other words, there is a unitary that must be applied to advance from gradient component $i$ to $j$, otherwise there will be a measurement that produces the incorrect result if no such unitary is applied. Fix $\{\Lambda, \mathbb{I} - \Lambda\}$ as the POVM elements of a gentle measurement. Assume without loss of generality that the outcome of measuring the gradient component with respect to a given state corresponds to the element $\Lambda = A^\dagger A$, and

$$\|S(\rho) - \rho\|_{\mathrm{tr}} \leq \alpha \tag{37}$$

where

$$S(\rho) = \frac{A\rho A^\dagger}{\mathrm{Tr}[\Lambda\rho]}.$$

Using a single copy of $\rho = |\psi\rangle\langle\psi|$ to extract all $M$ gradient components, requires advancing the state after measuring gently at each step, and thus, each measurement step must be on average $1/M$-gentle to ensure

$$\left\|S(U_M S(U_{M-1}...S(U_2 S(U_1 \rho U_1^\dagger)U_2^\dagger)...U_{M-1}^\dagger)U_M^\dagger) - \rho_M\right\|_{\mathrm{tr}} < 1, \tag{38}$$

where $\rho_M$ is the density matrix representation of the advanced gradient state $|\psi_M\rangle = U_M...U_2 U_1 |\psi\rangle$. If we allowed for any more damage at a particular step, we could eventually reach a point where subsequent measurements yield incorrect results, as the cumulative damage to the state may exceed 1. While the gentleness could be distributed across each gradient component in different ways, from the above lemma, we see that the more gentle the operator, the more trivial it becomes. Hence, if we had $(M-1)$ $0-$gentle measurements, they would be infinitely trivial and provide no information with 1 informative measurement. Hence, the least trivial set of measurements that achieve an average of $1/M$ gentleness would be to have each measurement be $1/M$ gentle. By Lemma (26), then each measurement will be $O(1/M)$-trivial, which implies

$$\mathrm{Tr}[\Lambda\rho_i] \leq e^{1/M} \mathrm{Tr}[\Lambda\rho_{i+1}]$$

for any two gradient states $\rho_i, \rho_{i+1}$. As $M$ increases, the estimates for all gradient components will converge. Therefore, the measurement operator has an exponentially vanishing dependence on the input states themselves and hence, provides little-to-no information about the gradient components. $\square$

## E.4 Multiple copies and non-collapsing measurements

Non-adaptive, non-collapsing measurements are, by assumption, measurements that do not disturb the state of a quantum system at all. Under this assumption, the complexity class, non-adaptive Collapse-free Quantum Polynomial time (naCQP) was introduced. With this ability, searching through an unstructured $M$-element list can be performed in $\tilde{O}(M^{\frac{1}{3}})$ time, which is faster than the optimal lower bound of $O(M^{\frac{1}{2}})$ given by Grover's search algorithm [Grover, 1996]. Importantly, time complexity in naCQP is measured as the number of oracle queries plus the number of non-collapsing measurements. This definition is considered more fitting, since any task in naCQP allows for exponentially many non-collapsing measurements to be made and should thus, be accounted for.

Interestingly, one may still violate Grover's bound by allowing for approximately non-collapsing measurements. First, note that

$$\|\rho - \rho'\|_{\mathrm{tr}} = 0$$

for non-collapsing measurements, where $\rho'$ is the normalized state after measurement. In the approximately non-collapsing regime, assume that a measurement operator can be applied to a tensor product of the state $\rho$ such that

$$\left\|\rho^{\otimes m} - \rho'^{\otimes m}\right\|_{\mathrm{tr}} \le \alpha.$$

As $\alpha \to 0$, we recover the non-collapsing measurement regime. In the gradient setting, approximately non-collapsing measurements are merely gentle measurements. This leads to the following.

**Proposition 28.** *A sufficiently gentle measurement used for gradient extraction can solve an unstructured search problem in $\tilde{O}(M^{\frac{1}{3}})$ time.*

*Proof.* Reformulating the gentle gradient task as a search problem, let $M = 2^n$. Consider the state

$$\sin((2i+1)\theta) |x\rangle |1\rangle + \cos((2i+1)\theta) \sum_{y \in \{0,1\}^n, y \neq x} 2^{-\frac{M-1}{2}} |y\rangle |0\rangle \tag{39}$$

after applying $i = M^{\frac{1}{3}}$ Grover iterations, where $|x\rangle$ is the marked state. The probability of measuring the marked state is $|\sin((2i+1)\theta)|^2 \approx 1/M^{\frac{1}{3}}$. Suppose we can create the state $|\psi\rangle^{\otimes m}$, where $m = O(\log(M))$ by using $M^{\frac{1}{3}} \log(M)$ Grover queries. By having access to multiple copies of $|\psi\rangle$, assume that one may implement a $1/M$-gentle measurement on the copies as required for gradient estimation. Then, the probability of observing the marked state after a single gentle measurement is $\log(M)/M^{\frac{1}{3}}$. By performing $M^{\frac{1}{3}}$ gentle measurements on the $\log(M)$ copies, the probability of obtaining the marked state at least once is greater than $1 - e^{-\log(M)} = 1 - \frac{1}{M}$, using only $\tilde{O}(M^{\frac{1}{3}})$ Grover oracle queries and $O(M^{\frac{1}{3}})$ partially non-collapsing measurements, and thus, runs in time $\tilde{O}(M^{\frac{1}{3}})$. $\qquad\square$

