# OpenReview forum: "On quantum backpropagation, information reuse, and cheating measurement collapse"
_NeurIPS.cc/2023/Conference — NeurIPS 2023 spotlight_

### Official Review · Reviewer_vTRX · 2023-07-05

**Soundness:** 3 good
**Presentation:** 3 good
**Contribution:** 3 good
**Rating:** 6
**Confidence:** 3

**Summary:**

This paper explores the efficiency of training parameterized quantum models, from the perspective of backpropagation scaling. By leveraging some recent developments in shadow tomography and accessing multiple copies of a quantum state, the authors propose an algorithm that matches backpropagation scaling in quantum resources and reduces additional classical computational costs. The results provide valuable insights into the reusability of quantum information and the results are potentially meaningful for the future of quantum machine learning.

**Strengths:**

- The paper investigates a timely and relevant topic in quantum machine learning, comparing the efficiency of training parameterized quantum models to classical neural networks.
- The authors leverage recent developments in shadow tomography, providing a novel approach to study a meaningful problem on quantum neural networks.
- The proposed algorithm matches backpropagation scaling in quantum resources and reduces classical auxiliary computational costs.
- The angle of this work to study quantum neural networks is novel.

**Weaknesses:**

- The primary analysis is limited to quantum neural networks based on variational quantum circuits, which restricts the scope of the paper as many other types of quantum neural networks exist.
- The application of the results to general quantum machine learning algorithms is not convincingly demonstrated.
- The paper lacks a clear and well-motivated example demonstrating the application of the proposed methods, making it difficult to assess its practical implications and usefulness.

**Questions:**

- Could the authors provide more insights into the practical implications of the results and its potential applications?
- How do the results of this work extend to other quantum neural networks?
- There are a certain number of existing works on the gradients of quantum neural networks. How does Proposition 7 advance the known works?
- What is the relationship between this work and the problem of the barren plateau?

**Limitations:**

NA.

---

> ### Author Rebuttal · Authors · 2023-08-09
>
> We are grateful for the positive feedback from the reviewer and thank them for taking the time to provide an insightful review. We seek to address the reviewer’s concerns and questions below and would be happy to have any further discussions on any points that may remain unclear:
>
> Practical implications and applications: Many of the current architectural approaches in classical deep learning stem from the efficient scaling of gradient computations in number of parameters.  If the scaling was quadratic in number of parameters, as is the case in current quantum algorithms, no billion parameter networks would be usable.  If large sizes are needed to cross into an effective overparameterized regime, even in the quantum case, this would be virtually impossible.  This has the potential to dramatically shift the preferred architectures from quantum neural networks to kernel methods, or specialized networks made for training, as is sometimes done classically.  In either case, there can be large shifts in design choices and successful algorithms from this small factor alone.
>
> Extension of results to other quantum neural networks: Our results are pretty general and encompass almost all architectures termed ``quantum neural networks’’. Equation 8 defines a QNN model and one may see by the general form of U(theta) that layers of parameterized rotations and any fixed operations may be applied.
>
> Application of proposition 7 to known results: One example application of the results of proposition 7, is that it allows the magnitude of all Pauli observables to be measured using only 2 copies at a time, with a logarithmic number of repetitions.  This allows the screening step for Adapt VQE to be done with only a log number of repetitions in the candidate pool of operators, which is an exponential improvement over recently proposed schemes.  If more copies can be kept in quantum memory, this can be used to determine the signs as well, but this is not a requirement for improving current Adapt VQE operator pool schemes.
>
> Relationship with barren plateaus: The relationship between our work and the barren plateaus result is somewhat orthogonal. While the findings of both works indicate a difficulty in training quantum circuits, we examine the scaling of resources needed to estimate gradients to within some fixed precision. If one experiences a barren plateau where gradients are exponentially vanishing, a higher precision would be needed to extract enough information to update gradients.
>
> We thank the reviewer again for providing us with useful feedback and hope that these comments helped clarify the questions posed by the reviewer.

---

> > ### Comment · Reviewer_vTRX · 2023-08-16
> >
> > I thank the authors for their reply, which helps me better understand and evaluate their results.

---

### Official Review · Reviewer_efhf · 2023-07-05

**Soundness:** 3 good
**Presentation:** 2 fair
**Contribution:** 3 good
**Rating:** 5
**Confidence:** 3

**Summary:**

The paper explores whether parameterized quantum models can achieve comparable training efficiency to classical neural networks. From the perspective of reusing quantum information, the paper demonstrates that achieving backpropagation scaling in quantum models is not feasible without access to multiple copies of a state. With access to multi-copies assumption, the authors propose an algorithm that achieves backpropagation scaling using gentle measurement and online learning while reducing classical auxiliary computational costs. These findings shed light on reusing quantum information for the challenges of training large quantum models.


**Strengths:**

The paper investigates the backpropagation in quantum models which is interesting and of general interest to the community of QML. It combines online learning and shadow tomography to achieve $O(polylog(M))$ sample complexity for gradient estimation.

**Weaknesses:**

1. Even though the proposed method achieves $O(polylog(M))$ of sample complexity, it also requires exponential classical resources which is not practical for handling a large system.
2. It only provides the theoretical analysis and does not give some proof-of-principle numerics.
3. Some necessary details and the related brief introduction of the proposed methods should be listed in the manuscript instead of supplementary.


**Questions:**

1. In proposition 7(193), as the variational model is defined as the trace of a quantum state, i.e. $tr[U_\theta \rho U_\theta^\dagger]$, the loss will always be constant 1, so no matter what $\theta$ we choose the gradient will always be 0, so it is confusing that what's the contribution of the gradient in such setting and whether in the general case, it also achieves $O(\frac{\log(M)}{\epsilon^4})$ backpropagation scaling.
2. In the proof of theorem 12(281), As in definition 8, $\mathcal{U}(\theta)=e^{-i\theta_M P_M}\dots U_1$, why each parameter $\theta_i, i\in[M]$ associated with the same $P_i=Y_0\otimes Z_1$ and whether the first term in the Pauli string $P_i$ should not be $X_0$ when observable set as $Z_0$?
3. when we choose random Pauli strings $P_j$ and the initial setting of $\theta$ is NOT 0, whether the theorem 12 still holds?


**Limitations:**

---

> ### Author Rebuttal · Authors · 2023-08-09
>
> We would like to thank the reviewer for taking the time to go through the manuscript and provide useful feedback and suggestions. We are glad that the reviewer sees our work as relevant for the quantum machine learning community and we have drafted a response to the reviewer’s questions below:
>
> Regarding proposition 7 - We thank the reviewer for their careful reading of the paper.  Indeed the listed formula is incorrect and was meant to be updated to a form similar to that in Equation (8), using an ancilla qubit to correctly form the reduction to the measurement of all Pauli operators.  This will be corrected in all updated versions of the draft, but no essential results in the paper change from altering this example case.
>
> Question about the proof of theorem 12(281): It is true that each theta_i is associated to the same Pauli generator Y_0 \otimes Z_1. However, note that the U_i layers are all different, and encode the observables of the shadow tomography problem we are reducing from.
>
> If theta is not 0, does theorem 12 still holds: The main aim of this argument is to show that estimating gradients is a strictly harder task than performing shadow tomography. We show the reduction holds even for the special case of initialization to zero, which we believe makes our case even stronger. Also note that the layers U_i can be interpreted as consisting of gates where theta are not zero.
>
> We hope that these comments are helpful for the reviewer and we would be happy to provide any further details in the discussion period. We thank the reviewer again for taking the time to review our work and considering it for Neurips 2023.

---

> > ### Comment · Reviewer_efhf · 2023-08-20
> >
> > Thank you for the response, I have no further questions.

---

### Official Review · Reviewer_mok7 · 2023-07-05

**Soundness:** 3 good
**Presentation:** 2 fair
**Contribution:** 2 fair
**Rating:** 6
**Confidence:** 3

**Summary:**

The paper studies the scaling of computing the gradient of a quantum neural network. While in the classical case we can use backpropagation, which gives the same linear scaling for computing the gradient and the forward pass, in the quantum case, we would naively have to run a circuit for each component of the gradient, leading to a squared complexity in the number of parameters, which prevents studying quantum models with large number $M$ of parameters.
The authors formulate this problem in the language of shadow tomography and apply ideas from that field to the problem at hand.
This shows that while an $M\log M$ scaling is possible using polylog copies of the input state. This comes at a drawback of classical cost that scales as $2^n$ with $n$ the number of qubits. Resolving this exponential scaling would resolve some open problems in shadow tomography.

**Strengths:**

- Relevant problem in quantum ML.
- Connection with shadow tomography and application to scaling of gradient computation is new and can lead to new ways to think about the problem
- Rigorous statements supporting scaling
- Well written paper

**Weaknesses:**

- The paper relies on quantum information concepts that are not necessarily familiar with the ML audience at the conference.
- When talking about memory requirements of backprop in classical neural networks, one needs to store activations for reverse mode autodiff. This leads to memory that scales with the number of layers, while in the quantum case by analogy the number of qubits does not scale with the number of layers. The authors could comment about this.
- I was confused by Prop. 3: is the proof considering the case of a number of parameters $4^n$? I am not sure what we learn from this example since it does not seem to be part of the quantum neural networks we would like to train.
- The classical scaling as $2^n$ required for the proposed algorithm restricts a lot the class of problem for which this protocol can be useful.


**Questions:**

- Can you add a related work section to highlight the novelty with respect to previous work?
- Can you explain what is rotate/threshold check in figure 1? Can you add more intuition around proposition 7 to see how gentle measurements are used, e.g. what is alpha in this case? What is the role of $\sigma$?
- Can you comment on what problems could benefit from the proposed protocol, namely small $n$ and large $M$?
- Can you explain why approximations to $\sigma$ using for example tensor networks could be more robust than just simulating the quantum neural network with tensor network?

**Limitations:**

- As the authors say several time, the main limitation is that their algorithm come with a classical exponential scaling that limits its applicability.

---

> ### Author Rebuttal · Authors · 2023-08-09
>
> We highly appreciate that the reviewer regards our paper as well-written and technically sound, and we are grateful for their comments and insightful feedback. We aim to answer the questions raised below and would be happy to provide any further details as needed:
>
> Related work section: We would be happy to add a related work section as suggested by the reviewer, or to more explicitly point out the novelty of our contribution in the introduction. For readability, related work is currently referenced mainly in Remark 4, which demonstrates the current scaling for proposed gradient methods.  This comparison to related work helps highlight the need for improved scaling in many quantum gradient methods.
>
> Rotate/threshold check: In figure 1, the rotate blocks can be understood as sequentially applying parameterized operations in the quantum neural network in order to estimate the next gradient component. Intuitively, this is similar to sequential layers in a neural network. The threshold check uses sigma, the hypothesis state, to estimate gradient components and check whether these estimates are acceptable. It does so by performing a gentle operation using copies in memory that checks if the gradient estimate exceeds a particular threshold. If the threshold is not exceeded, the protocol advances to the next rotation and threshold check. If the threshold is exceeded, then a destructive measurement is performed and a fresh batch of copies are then needed to execute the rest of the protocol. A full description is given in the supplementary material.
>
> Intuition around Proposition 7: This proposition succeeds largely due to two factors, special properties of Paul operators and the ability to perform measurements on multiple copies. The procedure first exploits the fact that while two Pauli operators do not necessarily commute, i.e. P_1 P_2 \neq P_2 P_1, the tensor product P_1 \otimes P_1 does commute with P_2 \otimes P_2. With such an observation, and the ability to measure two copies, the procedure is able to estimate the magnitude of every gradient component with a number of copies that scales logarithmically in M, since measurements on two copies can be performed such that they are not completely destructive (i.e. alpha is small). Then, using the magnitude information, the procedure performs a so-called majority vote measurement to extract sign information. This too can be done in a gentle manner thanks to the promise induced by the magnitudes being a certain size (the smaller components are ignored).
>
> The role of sigma: Sigma is crucial to the success of our proposed algorithm, as well as other state-of-the-art shadow tomography protocols. It acts as a proxy for our true quantum state of interest and is used in the protocol to estimate gradients offline. The hypothesis state is typically refined as one conducts a particular protocol, such that it better approximates the true quantum state and we can then use it to obtain better estimates of the gradient. Refining sigma is the key to succeeding with quasi-linear scaling, as one can rigorously show that there will be a bounded number of updates necessary.  Without proof that sigma updates “quicky” we would be reduced to naive measurement collapse, as is shown in Appendix E to be inefficient.
>
> Problems that could benefit from the proposed protocol (small n, large M): This is a good question and an ongoing area of research. At present, there are proposed schemes to take advantage of quantum data (see for example, quantum advantage in learning from experiments) that see substantial advantages at a modest number of qubits ~20-40 using kernel schemes.  There the advantage results from quantum data availability, so classical computational simulation provides no competition.  Schemes utilizing parameterized quantum circuits may require a large number of parameters to learn from the same quantum states, and as such M may be in the 10’s of thousands or greater such that M >> n, in such cases it is not unreasonable to bear the burden of the classical computational expense incurred with current schemes to estimate the gradient.  In practice, quantum hardware is currently forced to keep n small, and there have been some empirical studies about overparameterizing such a model.   The empirical results of classical, overparameterized networks support the idea that best performance may lie beyond a phase transition in a large, overparameterized regime, and reaching such a regime is likely to require a nearly linear overall scaling.
>
> On tensor network approximations to σ - Our meaning here is that the requirements of shadow tomography are less stringent than more standard measures of accuracy, such as trace distance, which quantify worst case error on observables.  In the case of shadow tomography, it is only asked that most observables be correct, most of the time, colloquially speaking.  As a result, even hypothesis states that are strictly speaking, orthogonal to the true state, can satisfy this requirement and numerical approximations such as tensor network states may get many relevant observables correct.
>
> We thank the reviewer again for carefully reviewing our work and providing helpful feedback. If there are still any open questions for the reviewer, we will be happy to discuss and look forward to the upcoming discussion period.

---

> > ### Comment · Reviewer_mok7 · 2023-08-18
> >
> > Thank you for the rebuttal. I do not have further questions.

---

### Official Review · Reviewer_Hoi6 · 2023-07-06

**Soundness:** 4 excellent
**Presentation:** 3 good
**Contribution:** 4 excellent
**Rating:** 7
**Confidence:** 4

**Summary:**

The authors go over the backpropagation scheme for both classical and quantum machine learning methods. They also propose a novel quantum backpropagation algorithm based on quantum shadow tomography to reuse information and reduce the time complexity.

**Strengths:**

1. This paper provides a link between quantum backpropagation and quantum shadow tomography, which are both important in quantum computing.
2. This paper provides a thorough background check on quantum backpropagation, information reuse scheme in QST, and backpropagation scaling problem.
3. This paper is technically sound.

**Weaknesses:**

1. This paper is more like a report paper than a research paper to me, since the main contribution is to discuss in detail how reusing information can benefit quantum backpropagation, and the proposed algorithm seems quite trivial.

**Questions:**

1. Whether this level of space complexity on the classical device is acceptable?
2. Could you be more explicit about and also highlight the potential impact of this paper on the quantum machine learning society?


**Limitations:**

The major limitation is whether this paper fits the scope of the research paper in NeurIPS.

---

> ### Author Rebuttal · Authors · 2023-08-09
>
> We thank the reviewer for their helpful comments on our work and are grateful for the positive feedback. Regarding the reviewer’s questions, we have the following responses:
>
> The proposed algorithm: While the proof that our algorithm works leans heavily on the existing machinery of shadow tomography, we believe the connection of shadow tomography to a dynamic program through rotations at each threshold check to be an interesting contribution.  A naive implementation of shadow tomography with the same observables does not achieve a quasi-linear scaling in M, even with exponential offline computational effort, and at a glance it is not obvious this scaling is achievable due to the sequential nature of the program.  This challenge is also highlighted by the fact that no other gradient method for parameterised quantum circuits achieves a quasi-linear scaling in M. This represents substantial progress in this regard, but of course, as the reviewer points out, the classical storage may be an inhibitor, which we leave as an open problem.
>
> Acceptable space complexity: We believe that the answer here depends on the context of what the ultimate goal is, and what resources one has available relative to the problem size. In general, however, this (classical) space cost is indeed prohibitive and we suggest that future research try to address this issue by finding more efficient offline representations of states.  For small quantum learning applications, e.g. a number of qubits < 50, the exponential offline representation may be acceptable due to data advantages in manipulating the state.  This is in contrast to seeking strict computational advantages.  However at larger sizes, an efficient computational representation is sure to be needed, and efficient approximations or special cases remain an interesting open direction.
>
> Impact: In traditional machine learning, the efficient scaling of backpropagation has a strong influence in the architecture of learning models.  Certainly if the scaling in number of parameters was quadratic, billion parameter deep networks would essentially impossible today.  Hence, what seems like a small scaling difference has historically had an immense impact on the models used, and we believe this could be true in the quantum case as well. Parameterized quantum models are very popular in the field of quantum machine learning and there seems to be some ignorance around the scaling of resources to train such models, naively appealing to the success of large classical networks without accounting for the fundamental scaling difference of all previous schemes. Additionally, there have been no proposals for gradient methods of parameterized circuits which achieve quasi-linear scaling in M. Bringing these issues to light is much needed and developing a more efficient gradient procedure is much needed. We believe our work addresses these issues and provides much needed intuition as to why parameterized quantum models struggle to exhibit favorable scaling like that seen in neural networks equipped with backpropagation.
>
> Relevance for Neurips: Interest in quantum machine learning has increased greatly over the years, with accompanying (and sometimes bold) claims that quantum computers will improve current machine learning methods in some manner. Many papers aiming to demonstrate a quantum advantage use parameterized quantum models and often ignore the problem of resource scaling as the model size increases. Establishing clear results in this regard, is important for the machine learning community as a whole to get a sense of issues prevalent in these commonly used quantum machine learning models and how they may impact proposed results. Thus, we believe our work is relevant to both quantum and classical machine learning researchers.
>
> Again, we would like to thank the reviewer for their insightful feedback and positive assessment of our work. We hope that these additional comments help clarify any open questions and we would happy to supply more details if needed.

---

### Comment · Area_Chair_BEyA · 2023-08-13
**lets start discussion**

Hi all,

Thanks for serving as the reviewers for this submission. As the authors have already provided their responses. Now let's start further discussion. Here is a to-do list:

(1) Please acknowledge the authors when you finish reading their responses.

(2) Please indicate whether you have any further questions for the authors such that they can continue to response.

(3) Please indicate whether you are willing to change the ratings.

best,
The AC

---

### Decision · Program_Chairs · 2023-09-21

**Decision:**

Accept (spotlight)

**Comment:**

This paper explores the efficiency of training parameterized quantum circuits, from the perspective of backpropagation scaling. By leveraging quantum shadow tomography, the authors propose an algorithm that matches backpropagation scaling in quantum resources. This paper provides valuable insights into quantum machine learning especially the reusability of information in the backpropagation.

This paper investigates a timely and relevant topic in quantum machine learning, the training efficiency of parameterized quantum models, from a novel perspective. The proposed algorithm leverages the recent development in quantum shadow tomography and also matches the backpropagation scaling in quantum resources to reduce the classical computational costs. The authors also provide rigorous theoretical analysis to support their statements and demonstrate a clear view of how to transfer the insights from classical neural networks to parameterized quantum models.

Several problems regarding the practical use, the exponential scaling, and the scope of application are raised by reviewers and are solved by the authors' rebuttal.